# Virtual library docking for cannabinoid-1 receptor agonists with reduced side effects

Tia A. Tummino [1,2,13], Christos Iliopoulos-Tsoutsouvas[3,13], Joao M. Braz [4,13], Evan S. O'Brien [5], Reed M. Stein[1,2], Veronica Craik [4], Ngan K. Tran[3], Suthakar Ganapathy[3], Fangyu Liu[1], Yuki Shiimura [5,6], Fei Tong [3], Thanh C. Ho[3], Dmytro S. Radchenko [7], Yurii S. Moroz [7,8,9], Sian Rodriguez Rosado[4], Karnika Bhardwaj[4], Jorge Benitez[4], Yongfeng Liu [10], Herthana Kandasamy[11], Claire Normand[11], Meriem Semache[11], Laurent Sabbagh [11], Isabella Glenn [1], John J. Irwin [1], Kaavya Krishna Kumar [5] ✉, Alexandros Makriyannis [3,12] ✉, Allan I. Basbaum [4] ✉ & Brian K. Shoichet [1] ✉

Virtual library docking can reveal unexpected chemotypes that complement the structures of biological targets. Seeking agonists for the cannabinoid-1 receptor (CB1R), we dock 74 million tangible molecules and prioritize 46 high ranking ones for de novo synthesis and testing. Nine are active by radioligand competition, a 20% hit-rate. Structure-based optimization of one of the most potent of these ($K_i$ = 0.7 μM) leads to '1350, a 0.95 nM ligand and a full CB1R agonist of $G_{i/o}$ signaling. A cryo-EM structure of '1350 in complex with CB1R-$G_{i1}$ confirms its predicted docked pose. The lead agonist is strongly analgesic in male mice, with a 2-20-fold therapeutic window over hypolocomotion, sedation, and catalepsy and no observable conditioned place preference. These findings suggest that unique cannabinoid chemotypes may disentangle characteristic cannabinoid side-effects from analgesia, supporting the further development of cannabinoids as pain therapeutics.

Although the therapeutic use of cannabinoids dates back to at least the 15th century[1,2], their use in modern therapy, for instance as analgesics, has been slowed by their sedative and mood-altering effects and by concerns over their reinforcing and addictive potential[3,4]. With changes in cannabis' legal status and efforts to reduce reliance on opioids for pain management has come a renewed interest in understanding both the endocannabinoid system and how to leverage it for

therapeutic development[5]. Areas of potential application of such therapeutics include anxiety[6], nausea[7], obesity[8], seizures[9], and pain[10], the latter of which is the focus of this study. Progress has been slowed by the physical properties of the cannabinoids themselves, which are often highly hydrophobic, by the challenges of the uncertain legal environment, and by the substantial adverse side effects often attending on the drugs, including sedation, psychotropic effects, and

[1]Department of Pharmaceutical Chemistry, University of California, San Francisco, San Francisco, CA 94158, USA. [2]Graduate Program in Pharmaceutical Sciences and Pharmacogenomics, University of California, San Francisco, San Francisco, CA 94158, USA. [3]Center for Drug Discovery and Department of Pharmaceutical Sciences, Northeastern University, Boston, MA 02115, USA. [4]Department of Anatomy, University of California, San Francisco, San Francisco, CA 94158, USA. [5]Department of Molecular and Cellular Physiology, Stanford University School of Medicine, Stanford, CA 94305, USA. [6]Division of Molecular Genetics, Institute of Life Science, Kurume University, Fukuoka, Japan. [7]Enamine Ltd., 67 Winston Churchill Street, Kyiv 02094, Ukraine. [8]National Taras Shevchenko University of Kyiv, 60 Volodymyrska Stree, Kyiv 01601, Ukraine. [9]Chemspace LLC, 85 Winston Churchill Street, Suite 1, Kyiv 02094, Ukraine. [10]National Institute of Mental Health Psychoactive Drug Screening Program (NIMH PDSP), School of Medicine, University of North Carolina at Chapel Hill School of Medicine, Chapel Hill, NC 27599, USA. [11]Domain Therapeutics North America Inc., Montréal, Québec H4S 1Z9, Canada. [12]Department of Chemistry and Chemical Biology, Northeastern University, Boston, MA 02115, USA. [13]These authors contributed equally: Tia A. Tummino, Christos Iliopoulos-Tsoutsouvas, Joao M. Braz. ✉e-mail: kaavyak@stanford.edu; a.makriyannis@northeastern.edu; allan.basbaum@ucsf.edu; bshoichet@gmail.com

concerns about reinforcement and addiction[3]. Indeed, a characteristic defining feature of cannabinoids is their "tetrad" of effects[11]: analgesia, hypothermia, catalepsy, and hypolocomotion, the latter three of which may be considered adverse drug reactions. Meanwhile, inconclusive results in human clinical trials[12] have led to uncertainty in the field as to the effectiveness of cannabinoids as therapeutics. Nevertheless, the strong interest in nonopioid analgesics, and the clear efficacy of cannabinoids in animal models of nociception[13] have maintained therapeutic interest in these targets.

The cannabinoid-1 and -2 receptors (CB1R and CB2R), both members of the lipid family of G protein coupled receptors (GPCRs), are the primary mediators of cannabinoid activity[14]. These related receptors are largely differentiated by their expression profiles, with CB1R being expressed throughout the nervous system[15] and body, and CB2R primarily expressed in peripheral immune cells[16], though the exact distribution of the latter is still a subject of debate[17]. Based on these expression profiles, and supported by animal studies[18–20], CB1R is thought to be the major target involved in the psychotropic and tetrad effects of cannabinoids, as well as their analgesic effects in tests of nociception[21], though because of the high similarity of the two receptors, and the peripheral distribution of CB2R, a role for the latter receptor often cannot be discounted without direct testing.

The determination of the structures of these receptors[22–28] affords the opportunity to use structure-based methods to find ligands with unique chemotypes. Recent structure-based docking of make-on-demand virtual libraries have discovered such molecules[29] for a range of targets, often with differential pharmacology and reduced side effects[30–39]. By extension, additional CB1R chemotypes emerging from a structure-based approach might address some of the liabilities of current cannabinoids.

In this work, we computationally dock a library of 74 million virtual but readily accessible molecules against human CB1R, revealing a range of different scaffolds with relatively favorable physical properties. Structure-based optimization results in agonists with low-nanomolar binding affinities. The lead agonist is a potent analgesic with pain-relieving activity at doses as low as 0.05 mg/kg. It has a two- to 20-fold separation between analgesia and hypolocomotion, sedation, and catalepsy, addressing multiple negative aspects of the tetrad and highlighting the utility of structure-based screens for identifying chemical scaffolds with differential pharmacology.

## Results

### Virtual library docking against CB1R

The CB1R orthosteric site is large and lipophilic, explaining the high molecular weight and hydrophobicity of many of its ligands (Supplementary Fig. 1). These physical properties, however, often present metabolic and solubility challenges[40]. To balance drug-likeness with the properties necessary to complement the CB1R site, we sought molecules from a 74-million molecule subset of the "lead-like" ZINC15 database[41] between 350 and 500 amu with calculated LogP (cLogP) of between 3 and 5. This range overlaps with known CB1R property space while retaining polarity and size advantages over many cannabinoids (Fig. 1b). Each molecule was docked in an average of 3.04 million poses (orientations x conformations), totaling 63 trillion sampled and scored complexes. Seeking a diverse set of candidates, the top-ranking 300,000 were clustered into 60,420 sets and the highest scoring member of each was filtered for topological dissimilarity to known CB1/CB2 receptor ligands in ChEMBL[42,43] using Tanimoto coefficient (Tc <0.38) comparisons of extended connectivity fingerprints up to four bonds (ECFP4)[44]. High-ranking compounds that did not resemble known ligands were filtered for potential polar interactions with S383[7.39] and H178[2.65] (superscripts denote Ballesteros-Weinstein nomenclature[45]; see Methods, Fig. 1a, Supplementary Table 1). The top-ranking 10,000 remaining molecules were visually evaluated[46] in UCSF Chimera[47] for features that are not included or are approximated

in the scoring function and chemoinformatic filters, such as angles and distances of hydrogen bonds, dihedral strain, and incorrect protomer or tautomerization. Ultimately, 60 were prioritized for de novo synthesis. Of these, 46 were successfully made and tested for CB1R activity. Consistent with the design of the library, the molecules were smaller and more polar than most existing cannabinoid ligands, skirting the edge of property-space that is suitable for the large and hydrophobic CB1R orthosteric pocket (Fig. 1b).

In single-point radioligand displacement experiments, nine of the 46 prioritized molecules displaced over 50% of the radioligand, a 20% hit-rate (Fig. 1c, d, Supplementary Table 1). The top four of these (ZINC537551486, ZINC1341460450, ZINC749087800, and ZINC518437019, referred to as '51486, '0450, '7800, and '7019, respectively, from here on) were then tested in full concentration-response. All four displaced the radioligand $^3$H-CP-55,940 with $K_i$ values ranging from 0.7 to 4 µM (Fig. 1e). Owing to coupling to the inhibitory $G_{\alpha i}$ G protein, functional efficacy experiments monitoring a decrease in forskolin (FSK) simulated cAMP were tested using hCB1-expressing cells, with '51486 and '0450 showing modest agonism (Supplementary Fig. 2a). Limited solubility prohibited testing at high enough concentrations to obtain accurate $EC_{50}$ values; fortunately, colloidal aggregation counter-screens showed no such activity below 10 µM (Supplementary Fig. 2b-e), suggesting that activity seen in binding and functional assays is not due to this confound[48]. Taken together, the nine actives explore a range of chemotypes topologically distinct from known CB1 ligands (Supplementary Table 1), with relatively favorable physical properties (Fig. 1b, d).

Although the identified ligands are chemically and physically distinct from established cannabinoids, their docked poses recapitulate the interactions of the known ligands but do so with different scaffold and recognition elements. All of the four most potent ligands docked to adopt a "C" shaped conformation characteristic of the experimentally observed geometries of MDMB-Fubinaca[25], AM11542, and AM841[23] bound to CB1R. Similarly, all four are predicted to hydrogen-bond with S383[7.39], a potency-determining interaction at CB1 receptors observed in nearly all agonist-bound ligand-receptor complexes[49,50]. Additionally, all four ligands are predicted to make secondary hydrogen bonds to H178[2.65], a feature thought to be important for potency as well as agonism of CB1R[50]. Largely, these polar interactions are made using unique hydrogen-bond acceptor groups, such as an oxazole, oxathiine, or pyridazinone. Other characteristic hydrophobic and aromatic stacking interactions are found throughout the ligands, including with F268[ECL2], W279[5.43], and F174[2.61], though again often using different aromatic groups than found in the known ligands (Fig. 1f). The two most potent hits ('51486 and '0450) further exhibit aromatic stacking and hydrophobic packing with the twin-toggle switch residues W356[6.48] and F200[3.36] which are important for receptor activation[51,52] and may explain their stronger agonism profiles compared to '7019 and '7800, though on-target potency may also play a role.

To optimize these initial ligands, molecules with ECFP4 Tcs ≥ 0.5 to the four actives were sought among a library of 12 billion tangible molecules using SmallWorld (NextMove Software, Cambridge UK). These analogs were built, docked, filtered, and selected using the same criteria as in the original docking campaign. Between 11 and 30 analogs were synthesized for each of the four scaffolds. Optimized analogs were found for three of the four initial hits, improving affinity by between 5 and 24-fold, with '7019 improving 5-fold to a $K_i$ of 87 nM, '0450 improving 24-fold to 163 nM, and '51486 improving 16-fold to 44 nM (Supplementary Table 2). Based on the improvements in affinity from the first round, only the '51486 series was progressed to second-round analoging.

In subsequent bespoke synthesis (i.e., out-of-library synthesis), the first-round 44 nM analog of '51486, '60154, was further optimized to Z8504214042 ('4042) with a $K_i$ of 1.9 nM, upon the addition of a

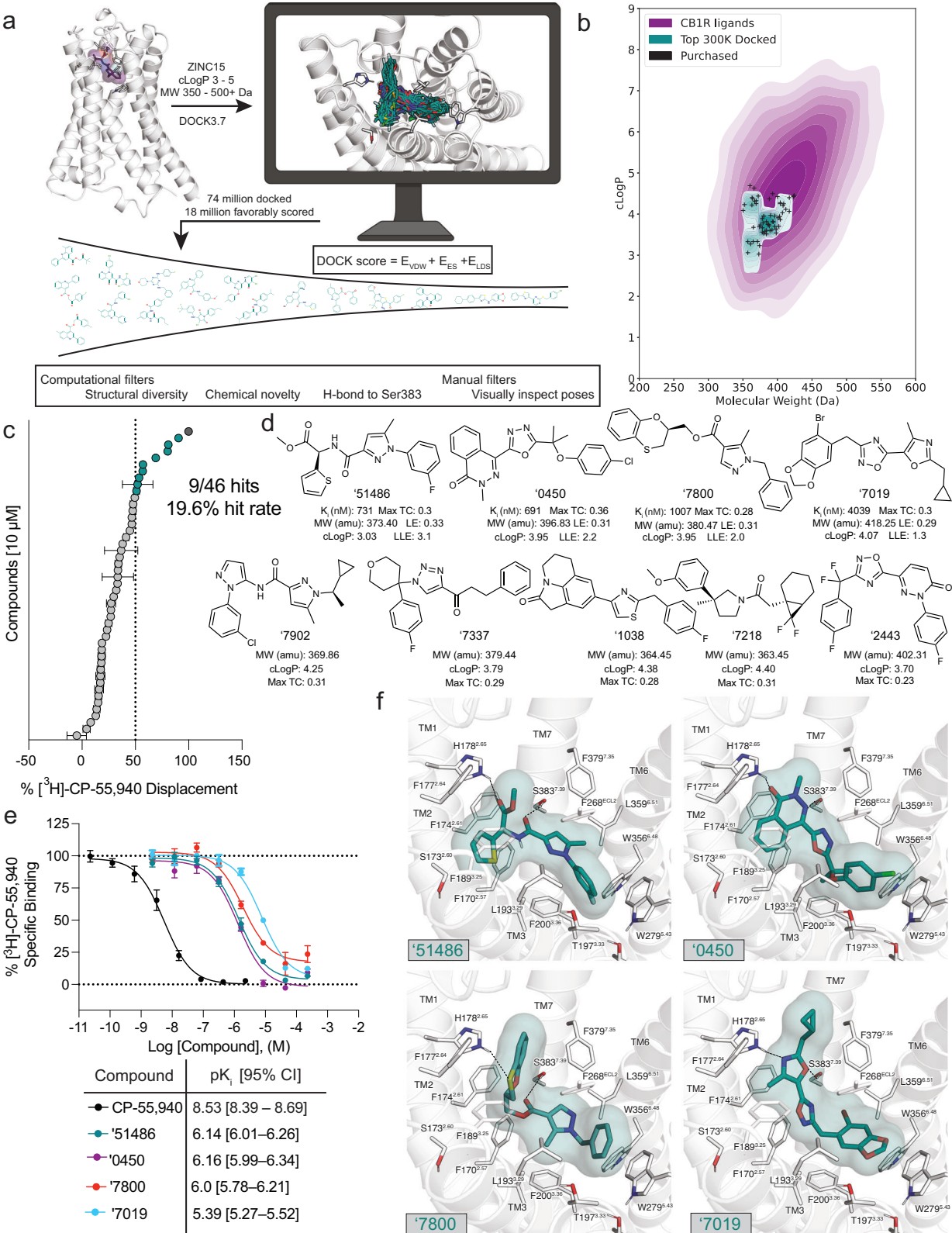

**Fig. 1 | Docking of a 74-million molecule library against the CB1R. a** Workflow of the docking campaign. **b** Overlap of physical properties of CB1R ligands versus the top docked and purchased ligands. **c** Single-point radioligand displacement data for the 46 tested compounds. **d** 2D structures and properties of the nine hits. **e** Secondary binding assay for the top four hits. **f** Docked poses of the top four hits with H-bonds and other binding pocket residues indicated. Data in **c** & **e** represent mean ± SEM from three independent experiments in triplicate. Created in BioRender. Stevens, J. (2025). https://BioRender.com/b34k743.

methyl group to the chiral center. As '4042 is a racemate, we purified it into it its component isomers, Z8526711350 ('1350) and Z8526708690 ('8690) using chiral chromatography (Supplementary Fig. 3, Supplementary Table 2) and measured CB1R binding to identify the active enantiomer. With $K_i$ values of 0.95 nM and 90 nM, respectively, '1350 was substantially more potent than its enantiomer, and subsequent functional studies revealed it to be the stronger agonist (Supplementary Fig. 3). In our effort to remove the chiral center and potential metabolic liabilities found in '1350, we substituted the methyl ester and thiophene of '1350 with difuranyl substituents, thus discovering Z8703004936 ('4936) with a $K_i$ of 7.5 nM. Figure 2 summarizes the structure-activity relationship (SAR) and docking models of the entire '51486 series.

Key learnings from the SAR include the importance of bulky and hydrophobic groups in the $R_1$ position, which is modeled to pack against W279$^{5.43}$ and T197$^{3.33}$ (Fig. 2c), with a meta-substituted $CF_3$ being most favorable. Substitutions of $R_2$ and $R_3$ had varying effects on ligand potency, with the most favorable functional groups including an ester/thiophene ('51486, '60154, '1350), difuran ('4936, '1090), or furan/phenyl ('5806, '1081) pairs. The ester carbonyl of '1350 and the furanyl oxygen of '4936 are modeled to hydrogen-bond with H178$^{2.65}$ of the receptor, though the distances suggest either water-mediated interactions or a weak hydrogen bond (Fig. 2d). In line with the docking model, the carboxylate analog of the ester, '4051, bound only weakly ($K_i = 5\,\mu M$, 5,000-fold less potent, Supplementary Fig. 4, Supplementary Table 2). Further, methylation of the chiral center ($R_4$ position, '1350, '4936, '5806) meaningfully improved affinity (approximately 6-fold in '4936, 15-fold in '5806, and 50-fold in '1350; Fig. 2e, Supplementary Fig. 4, and Supplementary Table 2). This addition is predicted to increase van der Waals interactions between the ligands and transmembrane helix 2. In contrast, methylation of the amide nitrogen at $R_5$ ('1066, '4388, and '1082) decreases the affinity of the scaffold by at least 100-fold and up to 2000-fold (Supplementary Fig. 4, Supplementary Table 2), despite the docking model predicting this to be an unsatisfied hydrogen bond donor. Though we did not try combining methylation of both the amide nitrogen and the chiral center (addition of N-Me to '1350, '4936, or '5806), we expect this too would negatively impact binding. Given the structural similarities and potency differences, '4051, '1066, and '4388 may be used as inactive probe pairs[53,54] in future research.

The leads that emerged, '1350 and '4936 are both potent binders of CB1R, with '1350 at 0.95 nM being 3-fold more potent ($P = 0.007$) and '4936 at 7 nM being 2.5-fold less potent ($P = 0.04$) than the widely used CB1R probe CP-55,940 (Fig. 2e, g, Supplementary Table 2). Although both '1350 and '4936 are more hydrophobic than the initial docking hit '51486, the lipophilic ligand efficiency (LLE; LLE = pIC50 − clogP) of '1350 improved from 3.1 to 4.7 (Fig. 2b), whereas '4936's LLE stayed approximately the same (3.2); both compare favorably to an LLE of 2.6 for the positive control CP-55,940 (cLogP = 5.66), which is more hydrophobic than either of the two docking-derived agonists.

## Agonism and subtype selectivity

Given the potent affinity of '1350 and '4936 (Fig. 2e) and several of their analogs (Supplementary Fig. 4, Supplementary Table 2), we next investigated their functional activity compared to the widely studied cannabinoid, CP-55,940[2]. We first measured $G_{i/o}$ mediated agonism via inhibition of forskolin-stimulated cAMP in the Lance Ultra cAMP assay (Methods). Both '1350, '4936 and their analogs are agonists in human CB1R-expressing cells (hCB1R), with $EC_{50}$ values commensurate with their affinities (Fig. 2f, g, Supplementary Fig. 4b, and Supplementary Table 2). Most molecules in this family are close to full agonists, with $E_{max}$ typically > 75%. The one exception is '4936, whose $E_{max}$ of 65% is more consistent with strong partial agonism (Fig. 2f, g, Supplementary Table 2). To verify that the activity is reproducible, we investigated it

further in orthogonal G protein, ß-arrestin-2, and off-target assays (Supplementary Fig. 3-7, Supplementary Tables 2-7).

Fortified by this potent activity, and to control for system bias[55–57], we investigated '1350 for differential recruitment of several G proteins and β-arrestin-2 ("signaling bias") against both CB1R and CB2R in the ebBRET bioSens-All® platform. We compared '1350's activity to CP-55,940 (Supplementary Fig. 5, Supplementary Tables 4-5, 7) by depicting the relative effectiveness[55] toward each signaling pathway (relative efficacy = $10^{\Delta log(Emax/EC50)}$, see Methods). In CB1R, '1350 was approximately 2-fold more relatively efficacious at recruiting $G_{i/o}$ and $G_{13}$ subtypes than was CP-55,940, though the pattern of effectors recruited was similar (Supplementary Fig. 5f-g). Notably, '1350's $E_{max}$ for $G_{13}$ and β-arrestin-2 was reduced, suggestive of partial rather than full agonism of these pathways (Supplementary Fig. 5g), which may have some physiological relevance. Differential activities for the highly related CB2R differed qualitatively (Supplementary Fig. 5; Supplementary Table 5, 7), with '1350 consistently being a less relatively efficacious partial agonist at CB2R (Supplementary Fig. 5h-i) compared to CP-55,940 across all recruited effectors. In summary, '1350 shows no strong functional selectivity or bias but is both more potent and relatively more efficacious than CP-55,940 at CB1R but not CB2R.

## Cryo-EM structure of the '1350-CB1R-$G_{i1}$ complex

To understand the SAR of '1350 at atomic resolution, and to template future optimization, we determined the cryo-EM structure of the '1350-CB1R-$G_{i1}$ complex (Fig. 3, Supplementary Fig. 8, see Methods) to a nominal resolution of 3.3 Å (Supplementary Table 8). Consistent with earlier structures of human CB1R in its activated state, the ligand occupies the orthosteric pocket formed by transmembrane helices (TMs) 2–3 and 5–7 and is capped by ECL 2 (Supplementary Fig. 9).

The experimental structure of '1350 superposes well on the docking-predicted pose of the R-enantiomer, which was the enantiomer with the better docking score to the receptor (-43 DOCK3.7 score versus -38 DOCK3.7 score for the S-enantiomer). In optical rotation studies, '1350 was shown to be the (-) enantiomer (see Supplementary Information file), together identifying '1350 as the R/(-) enantiomer. The predicted and experimental structures superposed with an all-atom RMSD of 0.78 Å (Fig. 3b). Despite the local resolution limit that prevents unambiguously modelling all rotamers, the density suggests that major interactions with CB1R predicted by the docking are likely preserved in the experimental structure, including the key hydrogen-bond between the amide carbonyl of the ligand and S383$^{7.39}$. The trifluoromethyl group is complemented by van der Waals and quadrupole interactions with residues W279$^{5.43}$ and T197$^{3.33}$, as anticipated by the docked structure, and consistent with the improvement in affinity by -1.7 kcal/mol ('51486 $K_i = 731$ nM vs. '60154 $K_i = 44$ nM, 17-fold increased $K_i$ from $CF_3$ addition only) on its replacement of the original fluorine.

## '1350 is analgesic with reduced cannabinoid side effects

**Off-target selectivity and pharmacokinetics.** Encouraged by their potency and signaling profiles, we progressed '1350 and '4936 into in vivo studies for pain relief. We began by investigating their selectivity against potential off-targets. '1350 and '4936 were tested for functional activity against a panel of 320 GPCRs at the PDSP (Supplementary Fig. 7g). Besides the expected activity at CB1R and CB2R, little activity was seen except for '4936 against ADORA-1. Intriguingly, no agonist activity was seen for the putative non-classical cannabinoid receptors GPR55, GPR18, or GPR119.

To minimize locomotor effects in pharmacokinetic exposure experiments, we used a dose of 0.2 mg/kg (Supplementary Fig. 10a-d). At this low dose, both '1350 and '4936 were found appreciably in brain and plasma, but not CSF compartments, with higher exposure in brain tissue ($AUC_{0\rightarrow inf} = 8180$ or 5610 ng*min/mL, respectively) than plasma ($AUC_{0\rightarrow inf} = 1510$ or 865 ng*min/mL, respectively). '1350 achieved

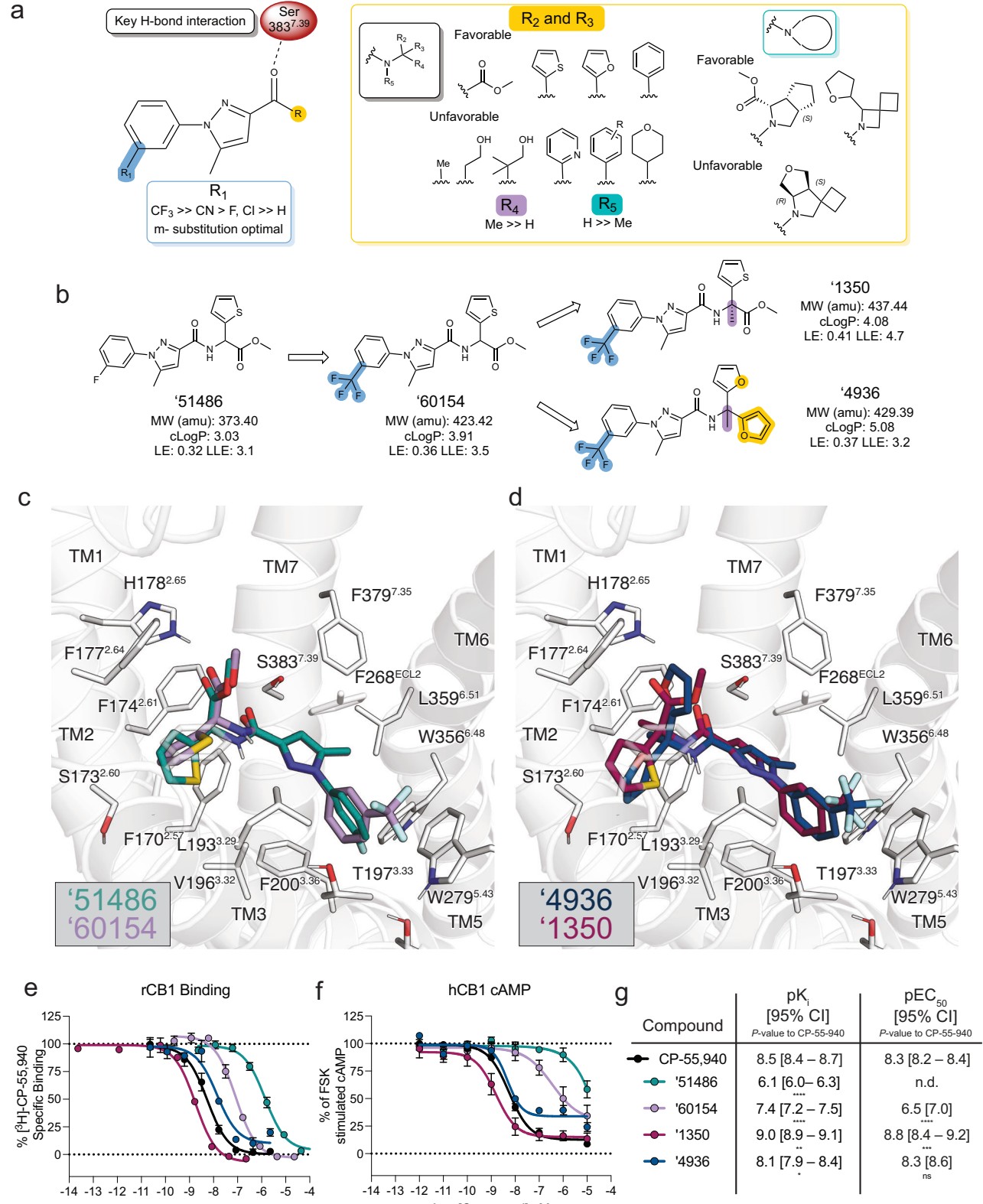

**Fig. 2 | Structure-activity relationships and optimization of '51486 to '1350 and '4936. a** Pharmacophore model based on the structure-activity relationships discovered via analoging '51486. **b** 2D structures and properties of the docking hit '51486 and most potent analogs. **c** Docking predicted pose of '51486 (teal) and '60154 (purple). **d** Docking predicted pose of '1350 (maroon) and '4936 (navy). **e**–**g** Binding affinity to rodent CB1R (rCB1) or functional cAMP inhibition to human CB1R (hCB1) by the '51486 analog series compared to CP-55,940. One-way ANOVA statistical significance of individual pKi (**e**; $F_{(6, 13)} = 152.2$, $P < 0.0001$) or pEC50 (**f**; $F_{(3, 19)} = 154.9$, $P < 0.0001$). Comparisons to CP-55,940 after correction with Dunnett's test of multiple hypotheses are depicted in the table. pKi: CP-55,940 vs. '51486 and '60154: $P < 0.0001$; vs. '1350: $P = 0.007$; vs. '4936: $P = 0.04$. pEC50: CP-55,940 vs. '60154: $P < 0.0001$; vs. '1350: $P = 0.0001$; vs. '4936: $P = 0.96$. Data in **e** & **f** represent mean ± SEM from two or three independent experiments run in technical triplicate, respectively. ns = not significant, *$P < 0.05$, **$P < 0.01$, ****$P < 0.001$.

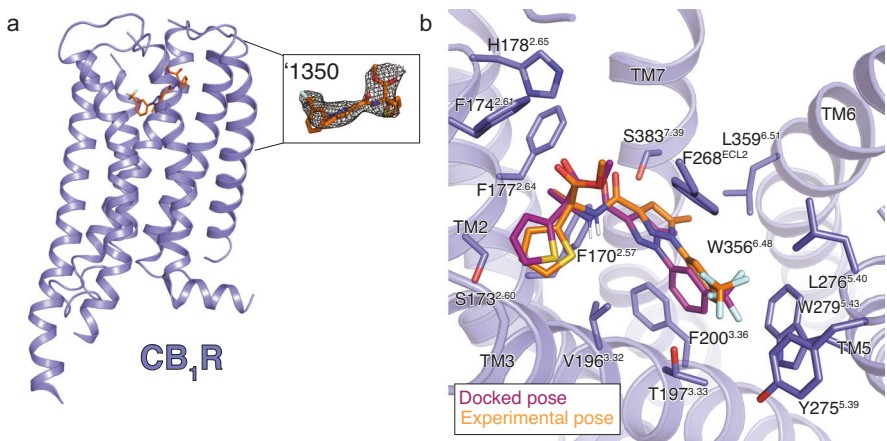

**Fig. 3 | Cryo-EM structure of '1350-CB1R-Gi1 complex. a** Cryo-EM model of '1350-CB1R highlighting the ligand density. **b** Overlay of the docked pose (maroon) with the experimental pose (orange) of '1350.

higher total concentrations in the brain ($C_{max}$ = 44.1 ng/g) and plasma ($C_{max}$ = 10.8 ng/mL or 25 nM) than '4936 (brain $C_{max}$ = 26.2 ng/g and plasma $C_{max}$ = 4.5 ng/mL or 11 nM) at this dose (0.2 mg/kg). A different pharmacokinetic profile was observed for the positive control CP-55,940 at 0.2 mg/kg compared to '1350 and '4936, reaching lower maximum concentrations in the brain ($C_{max}$ = 19.2 versus 44.1 and 26.2 ng/g), but with similar half-lives ($T_{1/2}$ = 127 min versus 112 and 125 min). One notable difference was seen in the plasma compartment, with a 5-10-fold increased $C_{max}$ for CP-55,940 compared to both '1350 and '4936.

To further explore '1350, we tested it for aqueous solubility (Supplementary Table 9), mouse plasma protein binding (Supplementary Table 10), mouse plasma and microsomal stability (Supplementary Fig. 10e-f, Supplementary Tables 11-12), membrane permeability, and P-glycoprotein (P-gp) activity (Supplementary Fig. 10g-i; see Methods). The molecule was soluble to 23 μM in PBS and was 94% plasma protein bound, values that are perhaps respectable for a lipid receptor ligand. Greater liabilities were seen in its relatively low stability in plasma (41-minute half-life) and liver microsomes (Supplementary Table 12). Conversely, '1350 was relatively membrane-permeable (Supplementary Fig. 10g) and was not a substrate of P-gp (Supplementary Fig. 10h). These observations are broadly in line with its physicochemical properties (cLogP = 4.08, cLogD = 4.50) and in vivo pharmacokinetics, where its plasma half-life is 111 min and where its level in the CSF is below the quantification limit, speaking to its low free fraction in the brain. Still, CP-55,940 is efficacious in vivo despite similar ($T_{1/2, brain}$ = 127 min) or in some cases worse (cLogP = 5.66, cLogD = 5.90, $C_{max}$ = 19.2 ng/g versus 44.1 ng/g) physicochemical and pharmacokinetic properties. Further, the metabolic and pharmacokinetic profiles for '1350, '4936, and CP-55,940 are in line with their moderate druglike central nervous system multiparameter optimization (MPO)[58] values, which are higher (3.3 and 3.7 for '1350 and '4936, respectively versus 3.0 for CP-55,940) for the docking-derived ligands (Supplementary Table 13), suggestive of moderate alignment with key CNS drug desirability properties. Taken together, '1350 has a modestly favorable pharmacokinetic profile and is therefore the focus of the proceeding efficacy experiments.

**Anti-allodynia and analgesia.** We tested the efficacy of '1350 in vivo in models of acute and chronic pain. We first focused on acute thermal pain. In tail flick, hot plate, and Hargreaves tests of thermal hypersensitivity, '1350 dose-dependently increased tail flick and paw withdrawal latencies. We recorded significant analgesia, namely latencies above baseline, at as little as 0.1 mg/kg dosed intraperitoneally (i.p.) (Fig. 4a, b, Supplementary Fig. 11a). We also recorded increased

latencies with the positive control CB1R ligand CP-55,940, but at higher doses (0.5 mg/kg for tail flick and hot plate, and 0.2 mg/kg in the Hargreaves test. Finally, the achiral analog of '1350, '4936, which substitutes the methyl ester and thiophene for difuran substituents was also tested in the hot plate assay (Fig. 4b), showing increased response latencies beginning at 0.5 mg/kg.

Next, we assessed the analgesic properties of '1350 in chronic pain models. As illustrated in Fig. 5a, 0.2 mg/kg i.p. of '1350 was also analgesic in the Complete Freund's Adjuvant (CFA)-induced inflammatory pain model, increasing paw withdrawal latencies to well-above pre-CFA baseline thresholds. Intriguingly, '1350, '4936, and CP-55,940 strongly reduced spared nerve injury-induced cold allodynia, a hallmark of neuropathic pain, significantly decreasing the combined total number of typical acetone-induced nocifensive behaviors, including paw withdrawals, shakes, and licks (Fig. 5b, Supplementary Fig. 11b). Finally, in the formalin model, an i.p. administration of 0.2 mg/kg '1350 profoundly decreased the duration of both phase 1 and phase 2 nocifensive behaviors throughout the 60-minute observation period (Fig. 5c). We conclude that the docking-derived CB1R agonists have therapeutic potential across multiple pain modalities in both acute and chronic pain settings.

**On target analgesic activity: CB1R vs. CB2R.** Because of the high sequence similarity of the CB1 and CB2 receptors, and the potential role of the latter in analgesia, we investigated the role of the two receptors in the analgesia of our lead. Consistent with CB1R being the target of '1350 in vivo, total knockout of CB1R in the mouse completely blocked the analgesic effect of '1350, but not of morphine, in the tail flick assay (Fig. 5d). Conversely, knockout of CB2R in the mouse did not decrease the analgesic effects of '1350 in the hot plate assay (Supplementary Fig. 11c). These observations suggest that the anti-allodynic, anti-hyperalgesic, and analgesic effects of '1350 are CB1R and not CB2R, dependent.

**Cannabinoid tetrad of behaviors.** The cannabinoid tetrad of behaviors is widely used to assess CNS engagement of cannabinoid receptors by ligands[11]. This suite of tests measures the four in vivo hallmarks of CB1R agonism, namely analgesia and three common cannabinoid side-effects—hypothermia, catalepsy, and hypolocomotion or sedation. We therefore examined our lead '1350 in this panel of potential side-effects.

**Reduced sedation at analgesic doses.** Hypolocomotion, one of the four features of the tetrad, is a commonly assessed proxy for the sedative side-effect of cannabinoids. Sedation is not only an important

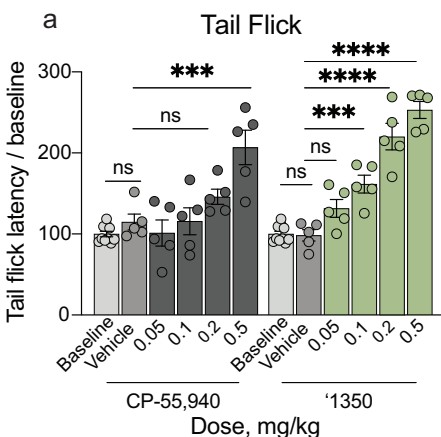

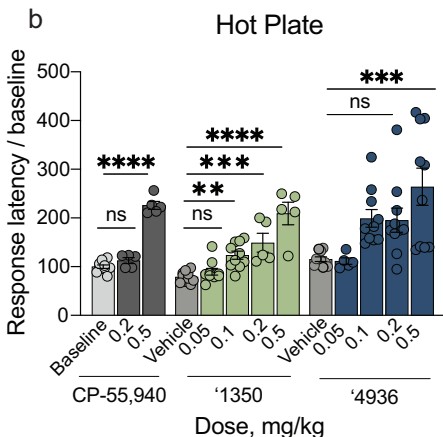

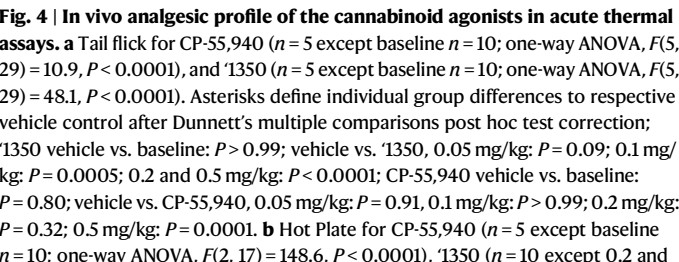

**Fig. 4 | In vivo analgesic profile of the cannabinoid agonists in acute thermal assays. a** Tail flick for CP-55,940 ($n = 5$ except baseline $n = 10$; one-way ANOVA, $F_{(5, 29)} = 10.9$, $P < 0.0001$), and '1350 ($n = 5$ except baseline $n = 10$; one-way ANOVA, $F_{(5, 29)} = 48.1$, $P < 0.0001$). Asterisks define individual group differences to respective vehicle control after Dunnett's multiple comparisons post hoc test correction; '1350 vehicle vs. baseline: $P > 0.99$; vehicle vs. '1350, 0.05 mg/kg: $P = 0.09$; 0.1 mg/kg: $P = 0.0005$; 0.2 and 0.5 mg/kg: $P < 0.0001$; CP-55,940 vehicle vs. baseline: $P = 0.80$; vehicle vs. CP-55,940, 0.05 mg/kg: $P = 0.91$, 0.1 mg/kg: $P > 0.99$; 0.2 mg/kg: $P = 0.32$; 0.5 mg/kg: $P = 0.0001$. **b** Hot Plate for CP-55,940 ($n = 5$ except baseline $n = 10$; one-way ANOVA, $F_{(2, 17)} = 148.6$, $P < 0.0001$), '1350 ($n = 10$ except 0.2 and 0.5 mg/kg $n = 5$; one-way ANOVA, $F_{(4, 35)} = 20.7$, $P < 0.0001$), and '4936 ($n = 10$ except 0.05 mg/kg $n = 5$; one-way ANOVA, $F_{(4, 40)} = 6.5$, $P = 0.0004$). Asterisks define individual group differences to baseline or vehicle after Dunnett's multiple comparisons post-hoc correction; vehicle vs. '1350, 0.05 mg/kg: $P = 0.85$; 0.1 mg/kg: $P = 0.006$; 0.2 mg/kg: $P = 0.0004$; 0.5 mg/kg: $P < 0.0001$; baseline vs. CP-55,940, 0.2 mg.kg: $P = 0.2$; 0.5 mg.kg: $P < 0.0001$; vehicle vs. '4936, 0.05 mg/kg: $P > 0.99$, 0.1 mg/kg: $P = 0.05$, 0.2 mg/kg: $P = 0.07$; 0.5 mg/kg: $P = 0.0002$. Data in **a** & **b** represent mean ± SEM. For all panels, $n$ denotes number of independent animals per group. ns = not significant, *$P < 0.05$, **$P < 0.01$, ****$P < 0.001$.

clinical adverse side effect of cannabinoids, but it also confounds preclinical reflex tests of analgesia, where unimpeded movement of a limb is the endpoint. Whereas '1350 showed locomotor deficits at 0.2 mg/kg in the open field test (Fig. 6a) and 0.5 mg/kg in the rotarod test (Fig. 6b), these effects occur at higher doses than do their analgesic effects, which occur at 2-10-fold lower doses, suggesting that hypolocomotion and sedation are not confounding the analgesic effects of '1350. Similarly, '4936 induces sedation in the rotarod only at the high dose of 2 mg/kg. Conversely, all analgesic doses tested with the positive control CP-55,940 caused motor impairment in both the rotarod and open field tests (Fig. 6a, b, d, e), suggesting that the analgesia produced by CP-55,940 is confounded by sedation at all doses (Figs. 4a, b, 5b, Supplementary Fig. 11a).

**Reduced catalepsy at analgesic doses.** To determine whether '1350 induced catalepsy, we measured the latency of compound-injected mice to move all four paws when placed on a vertical wire mesh. As expected, mice injected with the non-cannabinoid positive control haloperidol-induced catalepsy (Fig. 6c). Conversely, and consistent with the decreased sedative effects, analgesic doses (0.2 or 0.5 mg/kg) of '1350 did not induce cataleptic behavior post-injection. We did observe a non-significant, but nevertheless, increased latency to move the paws at a dose that also caused sedation (i.e., 1 mg/kg), suggesting that sedation may confound this measure. Knockout of CB1R eliminated this minimal cataleptic effect (Supplementary Fig. 11d). Unexpectedly, knockout of CB2R resulted in a statistically significant increase in catalepsy at the 1 mg/kg dose (Supplementary Fig. 11d) compared to wildtype. In contrast, '4936 does not induce catalepsy at any tested dose up to 1 mg/kg, whereas, CP-55,940 exhibited catalepsy starting at 0.2 mg/kg (Fig. 6c), consistent with the effects seen in the open field and rotarod tests (Fig. 6a, b). Here too, there was no window between analgesia and catalepsy for this widely used cannabinoid probe (Fig. 6f).

**'1350 induces hypothermia.** Finally, we examined the effect of '1350 on hypothermia. Here, we measured body temperature of mice implanted with telemetric probes continuously for 150 minutes. Both

CP-55,940 and '1350 induced hypothermia compared to baseline and vehicle (Fig. 7a), with '1350 showing increased hypothermia compared to CP-55,940, in contrast to the locomotor and cataleptic side-effect profiles.

Overall the leads discovered here, '1350 and '4936, have reduced adverse reactions at analgesic doses versus the classic cannabinoid CP-55,940. For the characteristic adverse tetrad behaviors, CP-55,940 induced meaningful catalepsy and sedation at the same concentrations where it conferred anti-allodynia and analgesia; for this widely used cannabinoid, it was impossible to deconvolute effects on pain from the adverse effects. This is as expected and is why the tetrad is considered characteristic of active cannabinoids. Conversely, depending on the nociceptive behavior, '1350 had up to a twenty-fold concentration window between anti-allodynia or analgesia versus catalepsy and sedation, and typically a five- to ten-fold window (Fig. 6d-f). This is most noticeable in the acetone test for cold allodynia, where '1350 demonstrated significant anti-allodynia at 0.05 mg/kg but only began to show increased latency to move paws suggestive of catalepsy at 1 mg/kg doses. In heat-based nociception, both in the tail-flick, which is reflex-based, and hot-plate, which is more affective, '1350 had at least a ten-fold window between anti-allodynia (significant at 0.1 mg/kg) and catalepsy (1 mg/kg highest tested dose) (Fig. 6f). In other behaviors the window dropped, for instance between heat-based responses in both the tail flick and hot plate and sedation as measured by the rotarod, it was only five-fold (Fig. 6e). However, in almost every behavior there was a meaningful window between nociception versus catalepsy and sedation, which is rare among cannabinoids such as CP-55,940. These findings are mimicked when testing '4936, where a 4–10-fold window was found between analgesia or anti-allodynia in the hot plate and acetone tests versus sedation measured by the rotarod. Notably, catalepsy was not measured for '4936, even at the highest tested non-sedating dose of 1 mg/kg, suggesting a therapeutic window greater than 5 for this behavior based on the minimal analgesic doses tested of this molecule.

**Pretreatment with '1350 increases the analgesic effect of morphine.** As cannabinoids have been shown to potentiate morphine

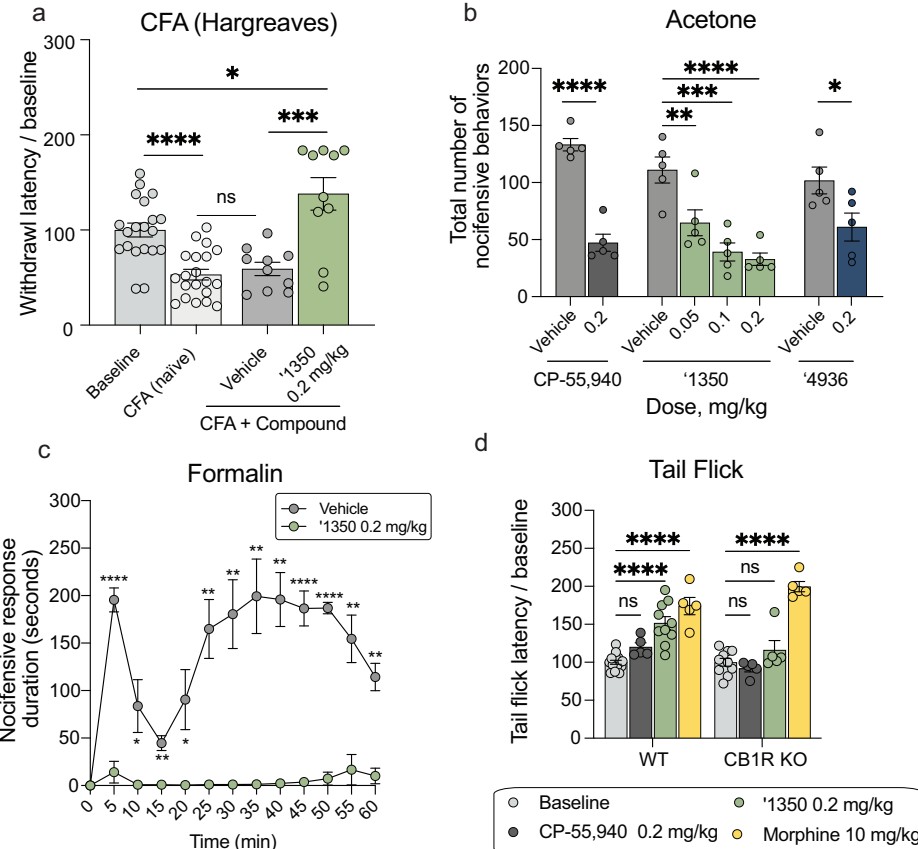

**Fig. 5 | Additional profiles of the cannabinoid agonists. a** Complete Freud's Adjuvant (CFA) test ($n = 10$ except baseline and CFA $n = 20$; two-tailed unpaired $t$-tests, '1350 vs. vehicle: $t(18) = 4.3$, $P = 0.0005$; '1350 vs. baseline: $t(28) = 2.4$, $P = 0.02$; CFA vs. vehicle: $t(28) = 0.6$, $P = 0.52$; CFA vs. baseline: $t(38) = 5.1$, $P < 0.0001$; asterisks define $t$-test $P$ value. **b** Acetone test (all $n = 5$; two-tailed unpaired $t$-tests, CP-55,940 vs. vehicle: $t(8) = 9.3$, $P < 0.0001$; '4936: $t(8) = 2.4$, $P = 0.04$; and '1350: one-way ANOVA, $F(3, 16) = 14.25$, $P < 0.0001$). For CP-55,940 and '4936, asterisks define $t$ test $P$ value. For '1350, asterisks define differences after Dunnett's multiple comparisons correction; vehicle vs. '1350, 0.05 mg/kg: $P = 0.0084$; 0.1 mg/kg: $P = 0.0002$; 0.2 mg/kg: $P < 0.0001$. **c** Formalin test (all $n = 5$; multiple two-tailed unpaired $t$-tests with Holm-Šídák correction; vehicle vs. '1350, 5 min: $P < 0.0001$; 10

and 20 min: $P = 0.03$; 15 min: $P = 0.003$; 25, 30, and 35 min: $P = 0.005$; 40 min: $P = 0.001$; 45 and 50 min: $P < 0.0001$; 55 min: $P = 0.005$; 60 min: $P = 0.002$. **d** Tail flick in wildtype (WT) versus CB1R knockout (KO) mice (WT: CP-55,940 and morphine $n = 5$, '1350 $n = 10$, baseline $n = 15$; CB1R KO: CP-55,940, morphine, and '1350 $n = 5$, baseline $n = 10$; two-way ANOVA; genotype x drug interaction: $F(4, 60) = 6.7$, $P = 0.002$; genotype: $F(1, 60) = 10.8$, $P = 0.002$; drug: $F(4, 60) = 45.5$, $P < 0.0001$). Asterisks define differences after Šídák's multiple comparisons correction; WT: baseline vs. CP-55,940: $P = 0.78$; baseline vs. '1350 and morphine: $P < 0.0001$; CB1R KO: baseline vs. CP-55,940: $P > 0.99$; vs. '1350: $P = 0.99$; vs. morphine: $P < 0.0001$. **a–d** represent mean ± SEM; $n$ denotes the number of independent animals per group. ns, not significant, $*P < 0.05$, $**P < 0.01$, $***P < 0.001$, $****P < 0.0001$.

analgesia[59–61], we investigated whether co-treatment of '1350 with morphine has better pain-relieving properties than morphine alone. Here, we combined low doses (0.05 and 0.1 mg/kg) of '1350 with morphine (3.0 mg/kg, i.p.) and tested the analgesic efficacy of the combination vs. morphine alone in the tail-flick assay. Mice co-injected with any combination of morphine and '1350 exhibited significantly longer tail-flick latencies than did mice injected with morphine alone (Fig. 7b). This result suggests that these combinations have at least an additive analgesic effect when combined, consistent with previous studies on circuitry[62] and CB1/2 R ligand polypharmacy with morphine[60–62].

**'1350 is not rewarding.** A major limiting factor in an analgesic's clinical utility, particularly opioids, is misuse potential because of rewarding properties. To determine whether '1350 exhibits such liabilities, we turned to the conditioned place preference (CPP) test in which mice learn to associate one chamber of the apparatus with a compound. If mice show a preference for the drug-paired chamber, then the compound is considered to be rewarding. As expected, mice injected with morphine significantly increased their preference for the chamber associated in which they received the drug versus its vehicle-

associated chamber (Supplementary Fig. 11e). In contrast, mice injected with '1350 spent comparable time in the '1350-paired and vehicle-paired chambers, indicating that '1350 does not induce preference at these doses. Similarly, we found that mice injected with the cannabinoid CP-55,940 did not spend more time in the drug-paired chamber; in fact, mice spent significantly more time in the vehicle-paired chamber, suggesting that CP-55,940 may actually induce some aversion, something not seen with '1350 but consistent with previous studies using a similar dose range for CP-55,940[63,64].

## Discussion

From a library of virtual molecules, structure-based docking has led to additional cannabinoid ligands that not only potently activate CB1R but are also strongly analgesic without key liabilities of classic cannabinoids. Three observations merit emphasis. First, from a tangible library of previously unsynthesized molecules, additional chemotypes for the CB1 receptor, physically distinct from previously known ligands, were found. Using structural complementarity, and the wide range of analogs afforded by the recently developed multi-billion molecule libraries, we optimized the best docking hit to a 0.95 nM $K_i$ and full agonist of CB1R-mediated $G_{i/o}$ signaling ('1350). Second, the

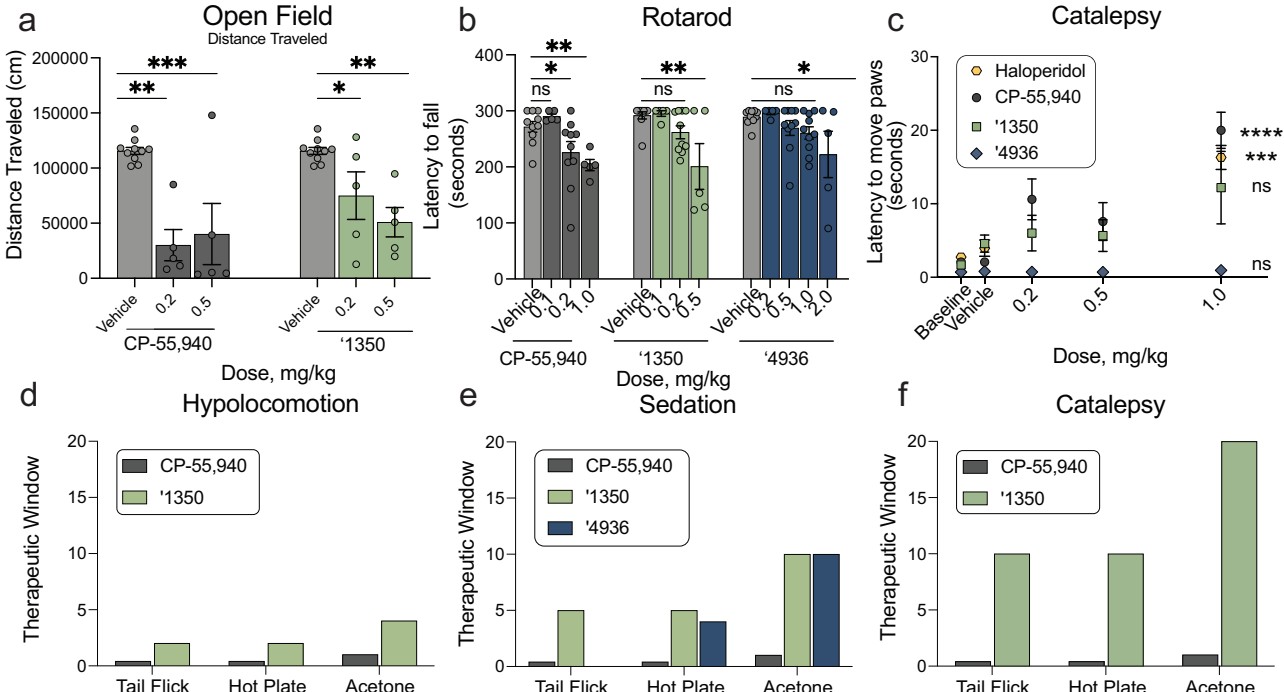

**Fig. 6 | Improved therapeutic window effects of '1350 and '4936. a** Open Field test ($n = 5$ except vehicle $n = 10$; one-way ANOVAs, '1350: $F_{(2, 17)} = 9.5$, $P = 0.002$; CP-55,940: $F_{(2, 17)} = 13.7$, $P = 0.003$). Asterisks define differences after Dunnett's multiple comparisons post-hoc correction; vehicle vs. '1350, 0.2 mg/kg: $P = 0.03$; 0.5 mg/kg: $P = 0.001$; vs. CP-55,940, 0.2 mg/kg: $P = 0.0006$; 0.5 mg/kg: $P = 0.002$. **b** Rotarod test ($n = 5$ for CP-55,940 0.1, 1.0 mg/kg, '1350 0.1, 0.5 mg/kg, '4936 0.2, 2.0 mg/kg. $n = 10$ for CP-55,940 vehicle, 0.2 mg/kg, '1350 vehicle, 0.2 mg/kg, and '4936 vehicle, 0.5, 1.0 mg/kg; one-way ANOVAs, CP-55,940: $F_{(3, 26)} = 5.7$, $P = 0.04$; '1350: $F_{(3, 26)} = 5.7$, $P = 0.004$; '4936: $F_{(4, 35)} = 2.7$, $P = 0.05$). Asterisks define differences after Dunnett's multiple comparisons post-hoc correction; vehicle vs. CP-55,940, 0.1 mg/kg: $P = 0.42$; 0.2 mg/kg: $P = 0.02$; 1.0 mg/kg: $P = 0.006$; vehicle vs. '1350, 0.1 mg/kg: $P = 0.99$; 0.2 mg/kg: $P = 0.32$; 0.5 mg/kg: $P = 0.002$; vehicle vs.

'4936, 0.2 mg/kg: $P = 0.99$; 0.5 mg/kg: $P = 0.71$; 1.0 mg/kg: $P = 0.4$; 2.0 mg/kg: $P = 0.02$. **c** Catalepsy test ($n = 5$ for haloperidol vehicle, 1.0 mg/kg, CP-55,940 0.2, 1.0 mg/kg, '1350 vehicle, 0.2, 1.0 mg/kg, and '4936 vehicle, 0.2, 0.5, 1.0 mg/kg. $n = 10$ for haloperidol baseline, CP-55,940 vehicle, 0.5 mg/kg. $n = 15$ for '1350 0.5 mg/kg. $n = 20$ for haloperidol, CP-55,940 baseline. $n = 30$ for '1350 baseline; one-way ANOVAs, CP-55,940: $F_{(3, 26)} = 10.7$, $P < 0.0001$; '1350: $F_{(3, 26)} = 1.03$, $P = 0.4$; '4936: $F_{(4, 29)} = 1.04$, $P = 0.4$; two-tailed unpaired $t$-test, haloperidol: $t_{(8)} = 6.2$, $P = 0.0002$). Asterisks define differences between 1 mg/kg compound dose to vehicle after $t$-test (haloperidol) or Dunnett's multiple comparisons post-hoc correction; '1350: $P = 0.29$; CP-55,940: $P < 0.0001$; '4936: $P = 0.67$. **d–f** Therapeutic windows. **a–c** represent mean ± SEM; $n$ denotes number of independent animals per group. ns, not significant, $*P < 0.05$, $**P < 0.01$, $***P < 0.001$, $****P < 0.0001$.

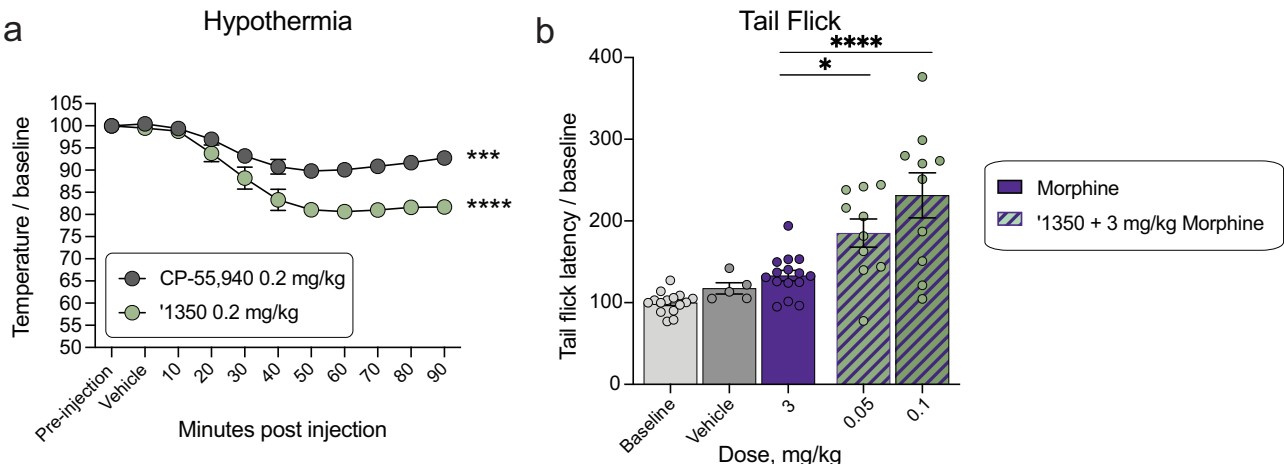

**Fig. 7 | Additional side effect and cotreatment profile of '1350. a** Body temperatures after treatment with CP-55,940 ($n = 5$; one-way ANOVA, $F_{(10, 44)} = 13.3$, $P < 0.0001$) and '1350 ($n = 3$; one-way ANOVA, $F_{(10, 22)} = 27.3$, $P < 0.0001$). Asterisks define differences between each group 90 min. post-dose to vehicle; '1350 0.2 mg/kg: $P < 0.0001$; CP-55,940 0.2 mg/kg: $P = 0.0005$. **b** Tail flick cotreatment of morphine with '1350 (morphine alone and baseline $n = 15$, '1350

plus morphine $n = 10$, vehicle $n = 5$, one-way ANOVA, $F_{(4, 50)} = 14.7$, $P < 0.0001$). Asterisks define cotreatment differences to morphine (3 mg/kg) after Dunnett's multiple comparisons post-hoc correction; '1350 0.05 mg/kg: $P = 0.029$; 0.1 mg/kg: $P < 0.0001$. All data represent mean ± SEM; $n$ denotes number of independent animals per group. ns, not significant, $*P < 0.05$, $**P < 0.01$, $***P < 0.001$, $****P < 0.0001$.

pose adopted by '1350 in a cryo-EM structure of its complex with CB1R-$G_i$ superposed closely on the docking prediction, explaining its SAR at atomic resolution and supporting future optimization. Third, '1350 is strongly anti-allodynic and analgesic across a panel of behavioral assays, and spares several of the characteristic adverse drug reactions of most cannabinoid analgesics, with a 2-20-fold window between analgesia and hypolocomotion, sedation, and catalepsy. These traits are unusual for cannabinoids, where sedation often closely tracks with analgesia and where catalepsy is among the tetrad of side-effects characteristic of cannabinoid agonists. Encouragingly, administration of morphine with low doses of '1350 show improved analgesia, suggesting that the combination of low doses of opioids and cannabinoids retains significant analgesia but potentially with a more favorable side effect profile, therefore expanding the therapeutic window of each compound on its own.

Several caveats bear mentioning. First, while our lead molecules are dissimilar to known cannabinoids by molecular fingerprint, they do share physical features with many of the characterized ligands, and even some core scaffold elements. For example, antagonists containing pyrazole-amides (e.g., rimonabant), which forms part of our most active series, are well-known. Second, the mechanistic bases for the disentanglement of sedation and catalepsy from analgesia remains uncertain. Often, clear differences in functional selectivity (signaling bias) or subtype selectivity explain the phenotypic differences among ligands[34,35,55,65]. Here, functional-selectivity differences between '1350, which features two reduced tetrad behaviors, and CP-55,940, which does not, were modest, with the only notable difference being the recruitment of $G_{13}$; the physiological effect of such a finding is not currently understood but should be explored in the future. Pronounced differences were, however, seen in the functional effects of '1350 on the CB1R and CB2R subtypes. Surprisingly, we observed an increase in catalepsy but no change in analgesia of '1350 in the CB2 knockout mice (Supplementary Fig. 11c-d). However, the mechanism underlying how partial agonism of CB2 would decrease cataleptic behaviors requires further validation. Taken together, we suspect that the separation of analgesic and other tetrad behaviors in the docking-derived molecules may reflect a combination of pharmacokinetic, pharmacodynamic, selectivity, and signaling, though without further investigation this remains speculative. For now, we can only lay the ability to disentangle analgesic efficacy from tetrad adverse reactions at the door of the chemotypes explored through the virtual libraries[66-68]. Whether the increased therapeutic window seen here in mice translates to higher-order species also remains to be explored. Finally, we note that while only agonists emerged from the optimization of the initial docking actives, these early docking hits spanned a wide range of chemotypes, and in early assays did not show strong agonism; we cannot rule out that some of them were ultimately antagonists, even though only agonists were sought. Docking, in our hands, remains better at finding ligands than making functional distinctions between them, such as predicting agonist or antagonist effects.

Despite these caveats, the main observations of this study seem clear. Employing synergistic computational and experimental approaches, including molecular docking, computational chemistry, medicinal chemistry, structural biology, in-depth molecular and in vivo pharmacology, cannabinoid-1 receptor analgesics with reduced in vivo side effects were discovered. A cryo-EM structure of the '1350-CB1-$G_{i1}$ complex confirmed its docking-predicted pose. The lead agonists are analgesic in several of behavioral assays, and unlike the control cannabinoid CP-55,940 have a 2-20-fold therapeutic window over hypolocomotion, sedation, and catalepsy. We suspect that additional chemotypes still remain to be discovered, and that these might further separate the dose-limiting side-effect aspects of the cannabinoid tetrad while maintaining analgesic potency, supporting the development of cannabinoid medicines to treat pain.

## Methods

### Ethical Compliance
Our research complies with all relevant ethical regulations. Pharmacokinetic experiments were performed by Bienta (Enamine Biology Services) in accordance with Enamine pharmacokinetic study protocols and Institutional Animal Care and Use Guidelines (protocol number 1-22/2020). Animal behavioral testing was approved by the UCSF Institutional Animal Care and Use Committee and were conducted in accordance with the NIH Guide for the Care and Use of Laboratory animals (protocol #AN195657).

### Molecular docking
A crystal structure of the active-state human CB1R receptor (PDB: 5XR8)[23] was used for docking calculations. As the goal was to find small-molecule, non-phytocannabinoid ligands, we used ligand coordinates from the cryogenic ligand MDMB-Fubinaca (PDB: 6N4B)[25], after overlaying the two receptor structures. The coordinates of Met363[6.55] were modified slightly, while maintaining the residue within the electron density to reduce a clash with the overlaid ligand indole group. The combined coordinates were minimized with Schrödinger's Maestro v.11.9 prior to the calculation of the docking energy potential grids. These grids were precalculated using CHEMGRID 3.2.1[69] for AMBER v.14[70] united atom van der Waals potential, QNIFFT v.22[71] for Poisson-Boltzmann-based electrostatic potentials, and SOLVMAP[72] for Generalized Born-derived context-dependent ligand desolvation. Atoms of the ligand determined in the cryo-EM structure (PDB: 6N4B), MDMB-Fubinaca, were used to seed the matching sphere calculation in the orthosteric site, with 45 total spheres used (these spheres act as pseudo-atoms defining favorable sub-sites on to which library molecules may be superposed[73]. The receptor structure was protonated using REDUCE v.2[74] and AMBER united atom charges were assigned[70]. Control calculations[46] using 324 known ligands extracted from the IUPHAR database[75], CHEMBL24[43], and ZINC15, and 14,929 property-matched decoys[76] were used to optimize docking parameters based on enrichment measured by logAUC[46], prioritization of neutral over charged molecules, and by the reproduction of expected and known binding modes of CB1R ligands. SPHGEN[73] was used to generate pseudo-atoms to define the extended low protein dielectric and desolvation region[30,77]. The protein low dielectric and desolvation regions were extended as previously described[78], based on control calculations, by a radius of 1.5 Å and 1.9 Å, respectively. The desolvation volume was removed around S383[7.39] and H178[2.65] to decrease the desolvation penalty near these residues and to increase the number of molecules that would form polar contacts with them.

A subset of 74 million large, relatively hydrophobic molecules from the ZINC15 database (http://zinc15.docking.org), with calculated octanol-water partition coefficients (cLogP, calculated using RDKit 2020.09.1: Open-source cheminformatics; http://www.rdkit.org) between 3 and 5 and with molecular mass from 350 Da to 500 Da, was docked against the CB1R orthosteric site using DOCK3.7.2[79]. Of these, more than 18 million were successfully fit. An average of 4706 orientations, and for each orientation, an average of 645 conformations was sampled. Overall, about 64 trillion complexes were sampled and scored. The total time was about 25,432 core hours, or less than 18 wall-clock hours on 1500 cores.

To reduce redundancy of the top-scoring docked molecules, the top 300,000 ranked molecules were clustered by ECFP4-based Tanimoto coefficient (Tc) of 0.5, and the best-scoring member was chosen as the cluster representative molecule. These 60,420 clusters were further filtered by calculating the Tc against >7000 CB1R and CB2R receptor ligands from the CHEMBL24[43] database. Molecules with Tc ≥ 0.36 to known CB1R/CB2R ligands were not pursued further.

After filtering for structural uniqueness, the docked poses of the best-scoring members of each cluster were filtered by the proximity of their polar moieties to Ser383[7.39], Thr201[3.37], or His178[2.65], and visually

inspected for favorable geometry and interactions. For the most favorable molecules, all members of its cluster were also inspected, and one of these was chosen to replace the cluster representative if they exhibited more favorable poses or chemical properties. Ultimately, 60 compounds were chosen for synthesis and testing.

## Make-on-demand synthesis and purity information

Of these 60, 52 were successfully synthesized by Enamine (an 87% fulfilment), but only 46 were ultimately screened due to poor DMSO solubility of six of the molecules. The purities of active molecules and analogs synthesized by Enamine were at least 90% and typically above 95%. The purity of compounds tested in vivo were >95% and typically above 98%. Synthetic routes[80], chemical characterization, and purity quality control information for a subset of hits can be found in the Supplementary Methods and a list of all tested molecules and their single point displacement data can be found in Supplementary Data 1. Molecules were drawn with CHEMDRAW21.0.0.

## Optical Rotation

Optical rotation values were measured at Enamine. The lead enantiomers ('1350 and '8690) and control enantiomers ('1066 and '6000) were tested using a MCP 300 polarimeter (Anton Paar) with a 50 mm cell at 21 °C and 589 nm (sodium D-line; c, $CH_3OH$). $[\alpha]_D$ values are given in $10^{-1}deg\ cm^2\ g^{-1}$.

## Ligand optimization

Analogs with Tcs $\geq$ 0.5 to the four most potent docking hits ('51486, '0450, '7800, and '7019) were queried in Arthor (v4.2.4) and Small-World (v 5.6.5; https://sw.docking.org, https://arthor.docking.org; NextMove Software, Cambridge UK) against 1.4 and 12 Billion tangible libraries, respectively, the latter primarily containing Enamine REAL Space compounds (https://enamine.net/compound-collections/real-compounds/real-space-navigator). Results were pooled, docked into the CB1R site, and filtered using the same criteria as the original screen. Between 11 and 30 analogs were synthesized for each of the four scaffolds. Second- and third-round analogs were designed in 2D space based on specific hypotheses and were synthesized at Enamine.

## MPO Calculations

Multiparameter optimization (MPO) values were calculated for our two lead molecules '1350 and '4936 and four control CB1R ligands- CP-55,940, WIN 55,212-2, MDMB-Fub, and Rimonabant. First, topological polar surface area (tPSA), negative log of the acid dissociation constant (pKa), and calculated log distribution coefficient at pH 7.4 (cLogD_{7.4}) were calculated using JChem's cxcalc command line tool (JChem-21.13.0, ChemAxon, https://www.chemaxon.com). These values, plus their molecular weights and cLogPs (calculated above) were put into a publicly available MPO calculator[58] to get the MPO scores.

## Radioligand Binding Experiments

The binding affinities of the compounds were obtained by competition binding using membrane preparations from rat brain (source of CB1; Bioivt, Cat. RAT00BRAINMZN) or HEK293 cells stably expressing human CB2R receptors (provided by the Laboratory of Ken Mackie) and [³H]-CP-55,940 as the radioligand, as described[81]. Briefly, membrane resuspended in TME containing 0.1% BSA (w/v) (TME-BSA) and equivalent to 25 µg of membrane protein was addedto each assay well. [³H]CP-55,940 was diluted in TME-BSA to yield final assay concentrations from an order of magnitude below to an order of magnitude above the ligand's Kd. Nonspecific binding was assessed in the presence of 5 µM unlabeled CP-55,940 for the saturation binding experiments. For competition binding experiments, the final concentration of [³H]CP-55,940 was 0.75 nM, with increasing concentrations of competitive ligand. All binding assays were performed at 30 °C for 1 h with gentle agitation. After

incubation, the samples were transferred to Unifilter GF/B-96-well filter plates, and unbound ligand was removed using a Packard Filtermate-196 cell harvester (PerkinElmer Packard, Shelton, CT). Filter plates were washed four times with ice-cold wash buffer (50 mM Tris-HCl and 5 mM MgCl 2 containing 0.5% BSA, pH 7.4), and bound radioactivity ywas quantified by liquid scintillation counting. Nonspecific binding was subtracted from total bound radioactivity to calculate specific radioligand binding (as pmol/mg membrane protein). The results were analyzed using nonlinear regression to determine the $IC_{50}$ and $K_i$ values for each ligand (Prism 9 by GraphPad Software, Inc., San Diego, CA). The $K_i$ values are expressed as the mean of two to three experiments each performed in triplicate.

## Functional assays

**Lance Ultra cAMP Accumulation Assay.** The inhibition of forskolin-stimulated cAMP accumulation assays was carried out using Perki-nElmer's Lance Ultra cAMP kit following the manufacturer's protocol. In brief, CHO cells stably expressing human CB1R (provided by the lab of Laura Bohn) were harvested by incubation with Versene (Thermo-Fisher Scientific, Waltham, MA) for 10 min, washed once with Hank's Balanced Salt Solution, and resuspended in stimulation buffer at -200 cells/µL density. The ligands at eight different concentrations (0.001-10,000 nM) in stimulation buffer (5 µL) containing forskolin (2 µM final concentration) were added to a 384-well plate followed by the cell suspension (5 µL; -1000 cells/well). The plate was incubated for 30 min at room temperature. Eu-cAMP tracer (5 µL) and Ulight-anti-cAMP (5 µL) working solutions were then added to each well, and the plate was incubated at room temperature for an additional 60 min. Results were measured on a Perkin-Elmer EnVision plate reader. The $EC_{50}$ values were determined by nonlinear regression analysis using Graphpad Prism 9 and are expressed as the mean of three experiments, each performed in triplicate.

**Cerep cAMP Inhibition Assay.** Compounds '4042 and '3737 were run through the Cerep HTRF cAMP assay for functional activity as agonists (catalog number 1744; Cerep, Eurofins Discovery Services; France). The hCB1/2 CHO-K1 cells (ATCC: CCL-61) are suspended in HBSS buffer (Invitrogen) complemented with 20 mM HEPES (pH 7.4), then distributed in microplates at a density of 5.103 cells/well in the presence of either of the following: HBSS (basal control), the reference agonist (stimulated control) or the test compounds. Thereafter, the adenylyl cyclase activator forskolin is added at a final concentration of 25 µM. Following 30 min incubation at 37 °C, the cells are lysed, and the fluorescence acceptor (D2-labeled cAMP) and fluorescence donor (anti-cAMP antibody labeled with europium cryptate) are added. After 60 min at room temperature, the fluorescence transfer is measured at λex = 337 nm and λem = 620 and 665 nm using a microplate reader (Envison, Perkin Elmer). The cAMP concentration is determined by dividing the signal measured at 665 nm by that measured at 620 nm (ratio). The results are expressed as a percent of the control response to a saturating concentration of CP-55,940, in this case the 10 nM datapoints. The $EC_{50}$ for the control CP-55,940 was 0.026 nM in the hCB1 assay and 0.082 nM in the hCB2 assay (data not shown). Each measurement was done in triplicate.

**Glosensor cAMP Accumulation Assay.** The GloSensor cAMP accumulation assay was performed as secondary validation assays (dose-response setup) as described in detail on the NIMH PDSP website at https://pdsp.unc.edu/pdspweb/content/PDSP%20Protocols%20II%202013-03-28.pdf. Briefly, hCB1-expressing cells were transfected with the GloSensor cAMP DNA construct overnight. On the day of assay, cells are removed from culture medium and receive 20 µl/well assay buffer, followed by addition of 10 µl of 3x drug solutions for 15 min at room temperature. To measure agonist activity for Gi-coupled receptors, 10 µl of 4 mM Luciferin supplemented with Isoproternol at final of

200 nM is added, and counting is done after 15 min. The results were analyzed using GraphPad Prism 9. Each experiment was performed in triplicate and functional $IC_{50}$ values were determined from the mean of three independent experiments.

**TRUPATH BRET2 $G_{oA}$ recruitment for CB2R.** CB2 receptor was co-expressed with. $G_{oA}$ dissociation BRET2 assays were performed as previously described with minor modifications[82]. In brief, HEK293T cells (ATCC: CRL-3216) were co-transfected overnight with human CB2 receptor, $G_{\alpha o}A$-Rluc, $G_{\beta 3}$, and $G_{\gamma 9}$-GFP2 constructs. After 18–24 h, the transfected cells were seeded into poly-L-lysine-coated 384-well white clear-bottom cell culture plates at a density of 15,000–20,000 cells and incubated with DMEM containing 1% dialyzed FBS, 100 U mL−1 of penicillin and 100 μg ml−1 of streptomycin for another 24 h. The next day, the medium was aspirated and washed once with 20 μL of assay buffer (1× HBSS, 20 mM HEPES, 0.1% BSA, pH 7.4). Then, 20 μL of drug buffer containing coelenterazine 400a (Nanolight Technology) at 5 μM final concentration was added to each well and incubated for 5 min, followed by the addition of 10 μL of 3X designated drug buffer for 5 min. Then, 10 μL of 4X final concentrations of ligands were added for 5 min. Finally, the plates were read in PHERAstar FSX (BMG Labtech) with a 410-nm (RLuc8-coelenterazine 400a) and a 515-nm (GFP2) emission filter, at 0.6-second integration times. BRET ratio was computed as the ratio of the GFP2 emission to RLuc8 emission. Data were normalized to percentage of CP-55,940 and analyzed in GraphPad Prism 9.1. Each experiment was performed in triplicate and functional $IC_{50}$ values were determined from the mean of four independent experiments.

**Tango β-arrestin-2 Recruitment Assay.** The Tango β-arrestin-2 recruitment assays were performed as described[83]. In brief, HTLA cells (a gift from the laboratory of R. Axel) were transiently transfected with human CB1R or CB2R Tango DNA construct overnight in DMEM supplemented with 10 % FBS, 100 μg ml−1 streptomycin and 100 U ml−1 penicillin. The transfected cells were then plated into poly-L-lysine-coated 384-well white clear-bottom cell culture plates in DMEM containing 1% dialysed FBS at a density of 10,000–15,000 cells per well. After incubation for 6 h, the plates were added with drug solutions prepared in DMEM containing 1% dialysed FBS for overnight incubation. On the day of assay, medium and drug solutions were removed and 20 μl per well of BrightGlo reagent (Promega) was added. The plates were further incubated for 20 min at room temperature and counted using the Wallac TriLux Microbeta counter (PerkinElmer). The results were analysed using GraphPad Prism 9. Each experiment was performed in triplicate and functional $IC_{50}$ values were determined from the mean of three independent experiments.

**DiscoverX PathHunter® β-arrestin-2 Recruitment Assay.** '4042 and '3737 were run through the PathHunter® β-arrestin-2 assay (catalog number 86-0001P-2070AG; DiscoverX, Eurofins Discovery Services; CA, USA). PathHunter cell lines (CHO-K1 purchased from ATCC lineage expressing hCB1) were expanded from freezer stocks according to standard procedures. Cells were seeded in a total volume of 20 μL into white walled, 384-well microplates and incubated at 37 °C for the appropriate time prior to testing. For agonist determination, cells were incubated with sample to induce response. Intermediate dilution of sample stocks was performed to generate 5X sample in assay buffer. 5 μL of 5X sample was added to cells and incubated at 37 °C or room temperature for 90 to 180 min. Vehicle concentration was 1%. Assay signal was generated through a single addition of 12.5 or 15 μL (50% v/v) of PathHunter Detection reagent cocktail, followed by a 1-hour incubation at room temperature. Microplates were read following signal generation with a PerkinElmer EnvisionTM instrument for chemiluminescent signal detection. Compound activity was analyzed using CBIS data analysis suite (ChemInnovation, CA). Percentage

activity was calculated using the following equation:

$$
\begin{aligned}
\% \text{ CP} &- 55,940 \text{ activity} \\
&= 100 \times \frac{(\text{mean RLU}_{\text{test sample}} - \text{mean RLU}_{\text{vehicle}})}{(\text{mean max}_{\text{CP}-55,940} - \text{mean RLU}_{\text{CP}-55,940})}
\end{aligned} \tag{1}
$$

The data were analyzed in GraphPad Prism 9.1 using "dose–response-stimulation log(agonist) versus response (four parameters)" and data were presented as $EC_{50}$ or $pEC_{50} \pm CIs$ of one independent experiment in duplicate.

**Signaling profiling of hCB1 and hCB2 using bioSensAll®.** ebBRET-based effector membrane translocation biosensor assays were conducted at Domain Therapeutics NA Inc. (Montreal, QC, Canada) as previously described[56]. CP-55,940 and test compounds were assayed for their effect on the signaling signature of the human cannabinoid receptor type 1 or 2 (hCB1 or hCB2) using the following bioSensAll® sensors: the heterotrimeric G protein activation sensors ($G_{\alpha s}$, $G_{\alpha i1}$, $G_{\alpha i2}$, $G_{\alpha oB}$, $G_{\alpha z}$, $G_{\alpha 13}$, $G_{\alpha q}$, $G_{\alpha 15}$) and the ßarrestin-2 plasma membrane (PM) recruitment sensor (in the presence of GRK2 overexpression). The HEK293 clonal cell line (HEK293SL cells) for bioSens-All experiments were derived and characterized previously[84] from HEK293 cells purchased from ATCC. Cells were maintained in Dulbecco's Modified Eagle Medium (DMEM) (Wisent) supplemented with 1% penicillin-streptomycin (Wisent) and 10% (or 2 % for transfection) fetal bovine serum (Wisent) at 37 °C with 5% CO2. All biosensor-coding plasmids and related information are the property of Domain Therapeutics NA Inc. The total amount of transfected DNA was adjusted and kept constant at 1 μg per mL of cell culture to be transfected using salmon sperm DNA (Invitrogen) as 'carrier' DNA, PEI (polyethylenimine 25 kDa linear, PolyScience) and DNA (3:1 ml PEI:mg DNA ratio) were first diluted separately in 150 mM NaCl then mixed and incubated for at least 20 min at room temperature to allow for the formation of DNA/PEI complexes. During the incubation, HEK293 cells were detached, counted, and re-suspended in maintenance medium to a 350,000 cells per mL density. At the end of the incubation period, the DNA/PEI mixture was added to the cells. Cells were finally distributed in 96-well plates (White Opaque 96-well /Microplates, Greiner) at a density of 35,000 cells per well. Forty-eight hours post-transfection, medium was aspirated and replaced with 100 μl of Hank's Balanced Salt Solution buffer (HBSS) (Wisent) per well using 450-Select TS Biotek plate washer. After 60 min incubation in this medium, 10 μL of 10 μM e-Coelenterazine Prolume Purple (Methoxy e-CTZ) (Nanolight) was added to each well for a final concentration of 1 μM immediately followed by addition of increasing concentrations of the test compounds to each well using the HP D300 digital dispenser (Tecan). All compounds were assayed at 22 concentrations with each biosensor after a 10-minute room temperature incubation period. BRET readings were collected with a 0.4 sec integration time on a Synergy NEO plate reader (BioTek Instruments, Inc., USA; filters: 400 nm/70 nm, 515 nm/20 nm). BRET signals were determined by calculating the ratio of light emitted by GFP-acceptor (515 nm) over light emitted by luciferase-donor (400 nm). All BRET ratios were standardized using the universal BRET (uBRET) equation:

$$
uBRET = \left( \frac{BRET \text{ ratio} - A}{B - A} \right) \times 10,000 \tag{2}
$$

where $A$ is the BRET ratio obtained from transfection of negative control and $B$ is the BRET ratio obtained from transfection of positive control. Data were normalized to the best fit values of CP-55,940 from each individual experiment before being pooled across replicates. If CP-55,940 had no response, data were left unnormalized and *uBRET* was used for plotting. The data were analyzed using the four-parameter logistic non-linear regression model in GraphPad Prism

9.1 and data were presented as means ± CIs of 1–4 independent experiments.

For relative efficacy calculations for '1350 and '4042 versus CP-55,940, first $E_{max}$ and $EC_{50}$ values were determined from dose-response curves to calculate the $\log(E_{max}/EC_{50})$ value for each pathway and each compound. Then, the difference between the $\log(E_{max}/EC_{50})$ values was calculated using the following equation:

$$\Delta \log\left(\frac{E_{max}}{EC_{50}}\right) = \log\left(\frac{E_{max}}{EC_{50}}\right)_{compound} - \log\left(\frac{E_{max}}{EC_{50}}\right)_{CP-55,940} \quad (3)$$

The SEM was calculated for the $log(E_{max}/EC_{50})$ ratios using the following equation:

$$SEM = \sigma/\sqrt{n} \quad (4)$$

where $\sigma$ is the standard deviation, and $n$ is the number of experiments.

The SEM was calculated for the $\Delta\log(E_{max}/EC_{50})$ ratios using the following equation:

$$SEM_{\left[\Delta \log\left(\frac{E_{max}}{EC_{50}}\right)\right]} = \sqrt{(SEM_{compound})^2 + (SEM_{CP-55,940})^2} \quad (5)$$

The compounds' efficacy toward each pathway, relative to CP-55,940, were finally calculated using the following equation:

$$\text{Relative Efficacy (RE)} = 10^{\Delta \log\left(\frac{E_{max}}{EC_{50}}\right)} \quad (6)$$

The relative efficacies were used in radar plots to demonstrate the relative compound effectiveness compared to CP-55,940.

Statistical analysis was performed using a two-tailed unpaired $t$-test on the $\Delta\log(E_{max}/EC_{50})$ ratios to make pairwise comparisons between tested compounds and CP-55,940 for a given pathway, where $P < 0.05$ was considered statistically significant.

**Signaling activity of PTHR and GSHR using the $G_s$ and $G_q$ bioSens-All® assays.** To demonstrate the ability of a receptor to couple to the $G_s$ and $G_q$ biosensors, HEK293 cells as described above were co-transfected with 200 ng of human PTH receptor (PTHR) or human Ghrelin receptor (GSHR) coding plasmids and plasmids coding for either $G_{\alpha s}$ or $G_{\alpha q}$ biosensor as indicated. Increasing amounts of PTH (for PTHR) or Ghrelin (for GSHR) were added to wells and BRET recorded 10 min later. Experimental data were produced in singleton and curves were fitted using the four-parameter logistic non-linear regression model (GraphPad Prism 9). Data are expressed as $u$BRET.

**Bimane Fluorescence.** A minimal cysteine version of CB1R was generated[85] where all the cysteine residues (except C256 and C264) were mutated to alanine. A cysteine residue was engineered at residue 336 (L6.28) on TM6, which was labeled with monobromobimane (bimane) by incubating 10 μM receptor with 10-molar excess of bimane at room temperature for one hour. Excess label was removed using size exclusion chromatography on a Superdex 200 10/300 Increase column in 20 mM HEPES pH 7.5, 100 mM NaCl and 0.01% MNG/0.001% CHS. Bimane-labeled CB1R at 0.1 mM was incubated with ligands (10 μM) for one hour at room temperature. Fluorescence data was collected at room temperature in a 150 μL cuvette with a Fluor-Essence v3.8 software on a Fluorolog instrument (Horiba) in photon-counting mode. Bimane fluorescence was measured by excitation at 370 nm with excitation and emission bandwidth passes of 4 nm. The emission spectra were recorded from 410 to 510 nm with 1 nm increment and 0.1 s integration time.

**GTP turnover assay.** Analysis of GTP turnover was performed by using a modified protocol of the GTPase-Glo™ assay (Promega) described previously[86]. Ligand-bound (10 μM ligand incubated for one hour at room temperature) or apo CB1R (1 μM) was mixed with G protein (1 μM) in 20 mM HEPES, pH 7.5, 50 mM NaCl, 0.01% L-MNG/0.001% CHS, 100 μM TCEP, 10 μM GDP and 10 μM GTP and incubated at room temperature. GTPase-Glo-reagent was added to the sample after incubation for 60 min ($G_{i1-3}$) and 20 min for ($G_o$). Luminescence was measured after the addition of detection reagent and incubation for 10 min at room temperature using a SpectraMax Paradigm plate reader.

**Colloidal Aggregation Counter-Screens**
**Dynamic Light Scattering (DLS).** Samples were prepared as 8-point half-log dilutions in filtered 50 mM KPi buffer, pH 7.0 with final DMSO concentration at 1% (v/v). Colloidal particle formation was measured using DynaPro Plate Reader II (Wyatt Technologies). All compounds were screened in triplicate.

**Enzyme Inhibition Counter-Screening Assays.** Enzyme inhibition assays to test for colloidal inhibition were performed at room temperature using CLARIOstar Plate Reader (BMG Labtech). Samples were prepared in 50 mM KPi buffer, pH 7.0 with final DMSO concentration at 1% (v/v). Compounds were incubated with 2 nM AmpC β-lactamase (AmpC) or Malate dehydrogenase (MDH) for 5 min. AmpC reactions were initiated by the addition of 50 μM CENTA chromogenic substrate (219475, Calbiochem). The change in absorbance was monitored at 405 nm for CENTA (219475, Calbiochem) or 490 for Nitrocefin (484400, Sigma Aldrich) for 60 sec. MDH reactions were initiated by the addition of 200 μM nicotinamide adenine dinucleotide (NADH) (54839, Sigma Aldrich) and 200 μM oxaloacetic acid (324427, Sigma Aldrich). The change in absorbance was monitored at 340 nm for 60 sec. Initial rates were divided by the DMSO control rate to determine % enzyme activity. Each compound was screened at 100μM in triplicate for three independent experiments, if enzyme inhibition greater than 30% was observed, 8-point half-log concentrations were performed in triplicate for three independent experiments. Data was analyzed using GraphPad Prism 9.1.

**Cryo-EM sample preparation and structure determination**
**Purification of hCB1.** hCB1R was expressed and purified as described previously[25]. Briefly, a N-terminal FLAG tag and C-terminal histidine tag was added to human full-length CB1. This CB1R construct was expressed in *Spodoptera frugiperda Sf9* insect cells with the baculovirus method (Expression Systems, Cat 94-001S). Insect cell pellets expressing CB1R was solubilized with buffer containing 1% lauryl maltose neopentyl glycol (L-MNG) and 0.1% cholesterol hemisuccinate (CHS) and purified by nickel-chelating Sepharose chromatography. The Ni column eluant was applied to a M1 anti-FLAG immunoaffinity resin. After washing to progressively decreasing concentration of L-MNG, the receptor was eluted in a buffer consisting of 20 mM HEPES pH 7.5, 150 mM NaCl, 0.05% L-MNG, 0.005% CHS, FLAG peptide and 5 mM EDTA. As the final purification step, CB1R was applied to a Superdex 200 10/300 gel filtration column (GE) in 20 mM HEPES pH 7.5, 150 mM NaCl, 0.02% L-MNG, 0.002% CHS. Ligand-free CB1R was concentrated to ~500 μM and stored in −80°C.

**Expression and purification of $G_{i/o}$ heterotrimer.** Expression and purification of all heterotrimeric G protein ($G_{i/o}$) follow similar protocols. Heterotrimeric $G_i$ was expressed and purified as previously described[87]. Wild-type human $G\alpha_{i1}$ subunit virus and wild-type human $\beta_1\gamma_2$ (with histidine tagged β subunit) virus were used to co-infect Insect (*Trichuplusia ni, Hi5*, Expression Systems, Cat 94011S) cells. Cells expressing the heterotrimeric, $G_i\beta_1\gamma_2$ G protein were lysed in hypotonic buffer and G protein was extracted in a buffer containing 1% sodium cholate and 0.05% n-dodecyl-β-D-maltoside (DDM, Anatrace). Detergent was exchanged from cholate/DDM to DDM on Ni Sepharose

column. The eluant from the Ni column was dialyzed overnight into 20 mM HEPES, pH 7.5, 100 mM sodium chloride, 0.1% DDM, 1 mM magnesium chloride, 100 µM TCEP and 10 µM GDP together with Human rhinovirus 3 C protease (3 C protease) to cleave off the His tag in the β subunit. 3 C protease was removed by Ni-chelating sepharose and the heterotrimeric G protein was further purified with MonoQ 10/100 GL column (GE Healthcare). Protein was bound to the column and washed in buffer A (20 mM HEPES, pH 7.5, 50 mM sodium chloride, 1 mM magnesium chloride, 0.05% DDM, 100 µM TCEP, and 10 µM GDP). The protein was eluted with a linear gradient of 0–50% buffer B (buffer A with 1 M NaCl). The collected G protein was dialyzed into 20 mM HEPES, pH 7.5, 100 mM sodium chloride, 1 mM magnesium chloride, 0.02% DDM, 100 µM TCEP, and 10 µM GDP. Protein was concentrated to about 200 µM and flash frozen until further use.

**Purification of scFv16.** scFv16 was purified with a hexahistidine-tag in the secreted form from Trichuplusia ni Hi5 insect cells using the baculoviral method. The supernatant from baculoviral infected cells was pH balanced and quenched with chelating agents and loaded onto Ni resin. After washing with 20 mM HEPES pH 7.5, 500 mM NaCl, and 20 mM imidazole, protein was eluted with 250 mM imidazole. Following dialysis with 3 C protease into a buffer consisting of 20 mM HEPES pH 7.5 and 100 mM NaCl, scFv16 was further purified by reloading over Ni a column. The collected flow-through was applied onto a Superdex 200 16/60 column and the peak fraction was collected, concentrated and flash frozen.

**CB1-G$_{i1}$ complex formation and purification.** CB1R in L-MNG was incubated with excess '1350 for ~1 h at room temperature. Simultaneously, G$_{i1}$ heterotrimer in DDM was incubated with 1% L-MNG/0.1% CHS at 4 °C. The '1350-bound CB1R was incubated with a 1.25 molar excess of detergent exchanged G$_i$ heterotrimer at room temperature for ~ 3 h. The complex sample was further incubated with apyrase for 1.5 h at 4 °C to stabilize a nucleotide-free complex. 2 mM CaCl$_2$ was added to the sample and purified by M1 anti-FLAG affinity chromatography. After washing to remove excess G protein and reduce detergents, the complex was eluted in 20 mM HEPES pH 7.5, 100 mM NaCl, 0.01% L-MNG/0.001% CHS, 0.0033% GDN/0.00033% CHS, 10 µM '1350, 5 mM EDTA, and FLAG peptide. The complex was supplemented with 100 µM TCEP and incubated with 2 molar excess of scFv16 overnight at 4 °C. Size exclusion chromatography (Superdex 200 10/300 Increase) was used to further purify the CB1-G$_i$-scFv16 complex. The complex in 20 mM HEPES pH 7.5, 100 mM NaCl, 10 µM **'1350**, 0.00075% L-MNG/0.000075% CHS and 0.00025% GDN/0.000025% CHS was concentrated to ~12 mg/mL for electron microscopy studies.

Cryo-EM data acquisition. Grids were prepared by applying 3 µL of purified CB1-G$_i$ complex at 12 mg/ml to glow-discharged holey carbon gold grids (Quantifoil R1.2/1.3, 200 mesh). The grids were blotted using a Vitrobot Mark IV (FEI) with 3 s blotting time and blot force 3 at 100% humidity at room temperature and plunge-frozen in liquid ethane. A total of 8324 movies were recorded on a Titan Krios electron microscope (Thermo Fisher Scientific- FEI) operating at 300 kV at a calibrated magnification of 96,000x corresponding to a pixel size of 0.8521 Å. Micrographs were recorded using a K3 Summit direct electron camera (Gatan Inc.) with a dose rate of 16.4 electrons/pixel/s. The total exposure time was 2.5 s with an accumulated dose of ~56.6 electrons per Å$^2$ and a total of 50 frames per micrograph. Automatic data acquisition was done using SerialEM.

**Image processing and 3D reconstructions.** Micrographs were subjected to beam-induced motion correction using MotionCor2[88] implemented in Relion 2.1.0[89]. CTF parameters for each micrograph were determined by CTFFIND4[90]. An initial set of 4,967,593 particle projections were extracted using semi-automated procedures and subjected to reference-free two-dimensional and multiple rounds of

three-dimensional classification in Relion 2.1.0[89] to remove low-resolution and otherwise poor-quality particles. From this step, 750,496 particle projections were selected for further processing in CryoSPARC (v.4.6.2)[91]. A final two-dimensional classification step to select for the highest-resolution particles resulted in a particle set containing 465,411 particles. These particles were reconstructed to a global nominal resolution of 3.3 Å (Supplementary Fig. 8) at FSC of 0.143 using non-uniform refinement. Further to improve the local resolution of the receptor only, we took Relion polished particles and performed non-uniform refinement in CryoSPARC, then we created a mask on the receptor alone using the fitted model. We performed local refinement with receptor alone mask, using pose/shift gaussian prior during alignment, 3° standard deviation of prior over rotation, 2 Å standard deviation of gaussian prior over shifts. Initial lowpass resolution was 6 Å with 5 extra final passes. This gave use better local resolution of the orthosteric pocket compared with non-uniform refinement alone (Supplementary Fig. 8). Finally, we built our model of the CB1 ligand based on the local refinement map. Local resolution was estimated within CryoSPARC[91]. A composite map was generated from these two maps in Phenix version 1.19.2.

Model building and refinement. The initial template of CB1R was the MDMB-Fubinaca-bound CB1-G$_i$ complex structure (PDB: 6N4B). Phenix.elbow was used to generate Agonist coordinates and geometry restrains. Models were docked into the EM density map using UCSF Chimera. Coot was used for iterative model building and the final model was subjected to global refinement and minimization in real space using phenix.real_space_refine in Phenix. Model geometry was evaluated using Molprobity. FSC curves were calculated between the resulting model and the half map used for refinement as well as between the resulting model and the other half map for cross-validation (Supplementary Fig. 8). The final refinement parameters are provided in Supplementary Table 8. The ligand symmetry accounted RMSD between the docked pose and cryo-EM pose of '1350 was calculated by the Hungarian algorithm in DOCK6.12[92].

### Off-target activity

**GPCRome.** Compounds '1350 and '4936 were tested at 3 µM for off-target activity against a panel of 320 non-olfactory GPCRs using PRESTO-Tango GPCRome arrestin-recruitment assay, as described above[83]. Receptors with at least three-fold increased relative luminescence over corresponding basal activity are potential hits. Screening was performed by the National Institutes of Mental Health Psychoactive Drug Screen Program (PDSP)[93]. Detailed experimental protocols are available on the NIMH PDSP website at https://pdsp.unc.edu/pdspweb/content/PDSP%20Protocols%20II%202013-03-28.pdf.

### Preclinical assessments

**Pharmacokinetics.** Plasma, brain, and CSF concentrations were measured for '1350, '4936, and CP-55,940 following a 0.2 mg/kg intraperitoneal (i.p.) dose. The batches of working formulations were prepared 5-10 min prior to the in vivo study. In each compound study, up to nine time points (5, 15, 30, 60, 120, 240, 360, 480 and 1440 min) were collected; each of the time point treatment groups included 3 male CD-1 mice. There was also a one mouse control group. All animals were fasted for 4 h before dosing. Mice were injected i.p. with 2,2,2-tribromoethanol at the dose of 150 mg/kg prior to drawing CSF and blood. Blood collection was performed from the orbital sinus in microtainers containing K$_3$EDTA. CSF was collected under a stereomicroscope from cisterna magna using 1 mL syringes. Animals were sacrificed by cervical dislocation after the blood samples collection. After this, right lobe brain samples were collected and weighted. All samples were immediately processed, flash-frozen and stored at -70 °C until subsequent analysis.

Plasma samples (40 µL) were mixed with 200 µL of internal standard solution. After mixing by pipetting and centrifuging for 4 min

at 5796 g, supernatant was injected into LC-MS/MS system. Solution of Difenoconazole (50 ng/ml in water-methanol mixture 1:9, v/v) was used as the internal standard (IS) for quantification of '1350, pirimiphos-methyl (200 ng/ml in water-methanol mixture, 1:4, v/v) for '4936, and mefenamic acid (100 ng/mL in water- acetonitrile mixture 1:9, v/v) for CP-55,940 were used as the IS for the quantifications. Brain samples were homogenized with 5 volumes of IS(80) solution using zirconium oxide beads (115 mg ± 5 mg) in The Bullet Blender® homogenizer for 30 s at speed 8. After this, the samples were centrifuged for 4 min at 20,817 g, and supernatant was injected into LC-MS/MS system. CSF samples (4 μL) were mixed with 100 μL of IS(80) solution. After mixing by pipetting and centrifuging for 4 min at 5,796 g, 1-6 μL of each supernatant was injected into LC-MS/MS system. The concentrations of the test compound below the lower limit of quantitation (LLOQ: 2-5 ng/mL for plasma and CSF, 1–5 ng/g for brain) were designated as zero. The pharmacokinetic data analysis was performed using noncompartmental, bolus injection or extravascular input analysis models in WinNonlin 5.2 (PharSight). Data below LLOQ were presented as missing to improve validity of $T_{1/2}$ calculations.

**Aqueous solubility.** The kinetic solubility assay was performed using a 20 mM stock solution of the experimental compound '1350 in 100% DMSO. Ondansetron was used as the positive control. Dilutions were prepared to a theoretical concentration of 400 μM in duplicates in phosphate-buffered saline pH 7.4 (138 mM NaCl, 2.7 mM KCl, 10 mM K-phosphate) with 2% final DMSO, were allowed to equilibrate at 25 °C on a thermostatic shaker for two hours and then filtered through HTS filter plates using a vacuum manifold. The filtrates of test compounds were diluted 2-fold with acetonitrile with 2% DMSO before measuring. To create the calibration curves, dilutions were prepared to theoretical concentrations of 0 μM (blank), 10 μM, 25 μM, 50 μM, 100 μM, and 200 μM in 50% acetonitrile/PBS with 2% final DMSO concentration. 200 μL of each sample was transferred to a 96-well plate and measured in the 230–550 nm range with a 5 nm step. The concentrations of compounds in PBS filtrate are calculated using a dedicated Microsoft Excel calculation script. Proper absorbance wavelengths for calculations are selected for each compound manually based on absorbance maximums (absolute absorbance unit values for the minimum and maximum concentration points within the 0–3 OD range).

**Plasma protein binding.** The assay was performed in duplicate in a multiple-use 96-well dialysis unit (HTD96b dialyzer), each with two chambers separated by a vertically aligned dialysis membrane of predetermined pore size (MWCO 14 kDa). 125 μL of non-diluted mouse plasma was spiked with either the positive control Verapamil or the experimental compound '1350 (1 μM, final DMSO and acetonitrile concentration was 0.005% and 1%, respectively). Compounds were added to one chamber and the same volume of PBS buffer pH 7.4 to the other chamber. A standard solution was created by mixing an aliquot of spiked plasma with blank buffer without dialysis. The test solutions (stability samples) and standard solutions (recovery samples) were incubated with shaking (10 g,) at 37 °C, 5% CO2, and a saturating humidity (~95%) for 5 h. The stability samples were immediately diluted with acetonitrile and stored at 4 °C until LC-MS/MS analysis. All samples were diluted 5-fold with 90% acetonitrile with internal standard with subsequent plasma proteins sedimentation by centrifuging at 5,796 g for 5 min. Supernatants were analyzed using HPLC system coupled with tandem mass spectrometer. The percentage of plasma protein-bound compound, recovery and stability were calculated using following equations:

$$\text{Protein Binding} = \left(1 - \frac{\text{area ratio in buffer}}{\text{area ratio in plasma}}\right) \times 100 \quad (7)$$

$$\text{Recovery} = \left(\frac{\text{area ratio in buffer} + \text{area ratio in plasma}}{\text{area ratio in stability sample}}\right) \times 100 \quad (8)$$

$$\text{Stability} = \left(\frac{\text{area ratio in recovery sample}}{\text{area ratio in stability sample}}\right) \times 100 \quad (9)$$

**Plasma stability.** Mouse plasma incubations were carried out in 5 aliquots of 60 μL each (one for each time point over 120 min), in duplicates. Compounds tested include the positive controls Verapamil, Propantheline, and test compound '1350 (1 μM, final DMSO concentration 0.005%). All compounds were incubated with shaking (10 g) at 37 °C, 5% CO$_2$ and a saturating humidity (~95%). The reactions were stopped by adding 240 μL 90% acetonitrile containing internal standard with subsequent plasma proteins sedimentation by centrifuging at 5796 g for 5 min. Supernatants were analyzed by the HPLC system coupled with a tandem mass spectrometer. The percentage of the test compounds remaining after incubation in plasma and their half-lives ($T_{1/2}$) were calculated.

**Hepatic microsomal stability.** Microsomal incubations were carried out in 96-well plates in 5 aliquots of 30 μL each (one for each time point). Liver microsomal incubation medium comprised of phosphate buffer (100 mM, pH 7.4), MgCl2 (3.3 mM), NADPH (3 mM), glucose-6-phosphate (5.3 mM), glucose-6-phosphate dehydrogenase (0.67 units/ml) with 0.42 mg of liver microsomal protein per ml. In the control reactions, the NADPH-cofactor system was substituted with phosphate buffer. Test compounds (2 μM, final acetonitrile concentration 1.6 %) were incubated with microsomes at 37 °C, shaking at 1.6 g. Five time points over 40 min were analyzed. The reactions were stopped by adding 5 volumes of acetonitrile with internal standard to incubation aliquots, followed by protein sedimentation by centrifuging at 4869 g for 5 min. Each reaction was performed in duplicates. Supernatants were analyzed using the HPLC system coupled with a tandem mass spectrometer. The elimination constant ($k_{el}$), half-life ($T_{1/2}$), and intrinsic clearance ($Cl_{int}$) were determined in a plot of ln(AUC) versus time, using linear regression analysis.

**Permeability assay.** Cultured canine MDR1 Knockout, Human MDR1 Knockin MDCKII (MDR1-MDCKII) (Sigma-Aldrich Cat#MTOX1303) cells were added (4 × 105 cells/mL) to each well of the HTS Multiwell Insert System and 25 mL of prewarmed complete medium was added to the feeder tray. To determine the rate of compounds transport in apical (A)-to-basolateral (B) direction, 300 μL of the test compound dissolved in transport buffer (9.5 g/L Hanks' BSS and 0.35 g/L NaHCO$_3$ with 0.81 mM MgSO$_4$, 1.26 mM CaCl$_2$, 25 mM HEPES, pH adjusted to 7.4) was added into the filter wells; 1000 μL of transport buffer was added to transport analysis plate wells. Ketoprofen, Atenolol, Quinidine and Digoxin were used as reference compounds. To determine transport rates in the basolateral (B) to apical (A) direction, 1000 μL of the test compound solutions were added into the wells of the transport analysis plate, the wells in the filter plate were filled with 300 μL of buffer (apical compartment). The effect of the inhibitor on the P-gp-mediated transport of the tested compounds was assessed by determining the bidirectional transport in the presence or absence of verapamil. The MDR1-MDCKII cells were preincubated for 30 min at 37° C with 10 μM of verapamil in both apical and basolateral compartments. After removal of the preincubation medium, the test compounds (final concentration 10 μM) with verapamil (10 μM) in transport buffer were added in donor wells, while the receiver wells were filled with the appropriate volume of transport buffer with 10 μM verapamil, respectively. The plates were incubated for 90 min at 37 °C under continuous shaking at 1.6 g, and aliquots were collected for LC-MS/MS analysis. All samples were mixed with 2 volumes of acetonitrile followed by protein sedimentation by centrifuging at 16097 g for 10 min.

Supernatants were analyzed using an HPLC system coupled with a tandem mass spectrometer.

**Animals.** Adult (8-10 weeks old) male C56BL/6 (strain # 664), CB1R knockout (strain #36108), and CB2R knockout (strain #5786) mice were purchased from the Jackson Laboratory. Sex was not considered in the study design and is not considered to significantly change study outcomes. Mice were housed in cages on a standard 12:12 h light/dark cycle with food and water ad libitum at 22 °C and relative humidity of 58%. Sample sizes were modelled on our previous studies and on studies using a similar approach, which were able to detect significant changes[94,95]. The animals were randomly assigned to treatment and control groups. Animals were initially placed into one cage and allowed to freely run for a few minutes. Then each animal was randomly picked up, injected with compound treatment or vehicle, and placed into a separate cylinder before the behavioral test.

**In vivo compound preparation.** Ligands were sourced from Enamine ('1350 and '4936) or Sigma-Aldrich (CP-55,940, Cat No. C1112; Haloperidol, Cat. No. H1512). '1350 and '4936 were resuspended in a 20% Kolliphor HS-15 (Sigma-Aldrich, Cat. No. 42966)/40% saline/40% water for injections (v/v/v) vehicle. CP-55,940 was resuspended in a 5% EtOH/5% Kolliphor-EL (Sigma-Aldrich Cat. No. C5135)/90% water for injections vehicle. Morphine (provided by the NIH) was resuspended in 100% saline. Haloperidol was resuspended in 20% cyclodextrin (Sigma-Aldrich, Cat. No. H107). All cannabinoid formulations were prepared in silanized glass vials.

**Behavioral analyses.** For all behavioral tests, the experimenter was always blind to treatment. Animals were first habituated for 30-60 min in Plexiglas cylinders and then tested 30 min after i.p. or i.t. injection of the compounds. The thermal (Hargreaves, hotplate and tail flick) and ambulatory (rotarod) tests were conducted as described[96]. Briefly, hindpaw thermal sensitivity was measured with a radiant heat source (Hargreaves) or on a hotplate at 52 °C. For the tail flick assay, sensitivity was measured by immersing the tail into a 50 °C water bath. For the ambulatory (rotarod) test, before testing with any compound, mice underwent three trainings on three consecutive days until they reach 300 sec. Each training has three sessions of five min each. Therapeutic window was calculated as the ratio of the minimum dose of side effect phenotype to the minimum dose of analgesic phenotype.

**CFA.** The CFA model of chronic inflammation was induced as described previously[97]. Briefly, CFA (Sigma) was diluted 1:1 with saline and vortexed for 30 min. When fully suspended, we injected 20 μL of CFA into one hindpaw. Heat thresholds were measured before the injection (baseline) and 3 days after the injection using the Hargreaves test.

**Open Field Test.** Thirty min after i.p. injection, mice were placed in the center of a round open-field (2 feet diameter) and their exploratory behavior recorded over the next 15 min. Distance traveled was used to represent open field behavior.

**Conditioned Place Preference.** To determine if '1350 was inherently rewarding we used the conditioned place paradigm as described[98]. Briefly, mice were first habituated to the test apparatus, twice, and their preference for each chamber recorded for 30 min (Pretest). Two conditioning days followed in which mice received the vehicle control or the compound, and 30 min later restricted for 30 min in the preferred or non-preferred chamber, respectively. On day 5 (Test day), mice were allowed to roam freely between the 3 chambers of the apparatus and their preference for each chamber recorded for 30 min. To calculate the CPP score, we subtracted the time spent in each chamber of the box on the Pretest day from that of the Test day (CPP score = Test - Pretest).

**Acetone Test.** Mice were placed on a wire mesh and thirty min after an i.p. injection of the compounds we applied a drop (50 μL) of acetone on the ventral aspect of the hindpaw, 5 times every 30 sec. We recorded the number of nocifensive behaviors (paw lifts/licks/shakes/bites) over the 5 applications.

**Formalin Test.** Thirty min after an i.p. injection of the compounds, mice received an intraplantar injection of a 20μL solution containing 2% formalin (Acros Organics) and we recorded the time mice spent licking/biting/guarding (nocifensive behaviors) the injected hindpaw over the next 60 min.

**Catalepsy Test.** Thirty min after an i.p. injection of the compounds, mice were placed on a vertical wire mesh and the latency to move all four paws was recorded.

**Body temperature measurements.** Body temperature (BT) was measured using a telemetric probe device (HD-X10; Data Science International). Briefly, under anesthesia, the probe device was placed in the mouse abdomen and a subcutaneous tunnel was created from the neck to the abdominal skin, through which a catheter (connected to the probe) was pulled and then inserted into the left carotid artery. Three weeks later, the implanted mice were singly housed in a cage that was placed on top of the DSI receiver (for probe signal detection). We monitored BT continuously over 2 h, in the following manner: 30 min (for baseline), 30 min after injection of the vehicle and then for 1 h after injection of the compound. Data were acquired using the Ponemah Telemetry acquisition software (DSI) and percent changes were presented relative to each mouse's baseline.

**Statistical analyses.** All statistical tests were run with GraphPad Prism 9.1 (GraphPad Software Inc., San Diego). Experiments of the compounds in the in vitro or in vivo assays were analyzed by unpaired two-tailed t-tests, one-way ANOVA, or two-way ANOVA, depending on the experimental design. If necessary for the statistical test, calculations were controlled for multiple hypothesis testing using a post-hoc test as described in the legends. Details of the analyses, including groups compared in post-hoc sets, number of animals per group, $t$ or $F$ statistics, $P$ values, definition of center, and dispersion and precision measures can be found in the legends.

### Reporting summary
Further information on research design is available in the Nature Portfolio Reporting Summary linked to this article.

## Data availability
Additional behavioral and pharmacological data are provided in the Supplementary Information file. Source Data are provided with this paper as Source Data file. The 1350-CB1R (receptor alone) model (Fig. 3) generated in this study has been deposited to the Protein Data Bank under accession code 9DGI and the map coordinates to the EMDB under accession code EMD-46828. The composite model was deposited to the Protein Data Bank under accession code 9EGO and the composite map to the EMDB under accession code EMD-47992. The '1350-CB1R-$G_{i/o}$ model was deposited to the Protein Data Bank under accession code 8GAG and the map was deposited to the EMDB under accession code EMD-29898. The Protein Data Bank entries for the structures used for the docking and model building can be found under accession code 5XR8 and 6N4B. The docking results, including DOCK scores, smiles, and ZINC IDs for all scored molecules and 3D poses for the top 500,000 molecules as well as input docking grids and optimized structures are available on the LSD website [https://lsd.docking.org/targets/CB1R]. Synthetic methods, chemical identities, purities (LC/MS), yields and spectroscopic analysis (H-NMR) for active compounds are provided in Supplementary Methods. A list of all de

novo compounds, their 2D structures, and their synthetic purities can be found in Supplementary Data 1. All compounds may be ordered from Enamine. Source data are provided with this paper.

## Code availability

DOCK3.7 is freely available for non-commercial research in both executable and code form (http://dock.compbio.ucsf.edu/DOCK3.7/). A web-based version is freely available to all (http://blaster.docking.org/). Usage instructions can be found at https://sites.google.com/site/dock37wiki. The library used here is freely available (http://zinc15.docking.org, http://zinc20.docking.org).

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

## Acknowledgements

This work is supported by DARPA grant HR0011-19-2-0020 (B.K.S., A.I.B., & J.J.I.), US NIH grant R35GM122481 (B.K.S.), US NIH grant R01GM71896 (J.J.I.), US R35NS097306 (A.I.B.), Open Philanthropy (A.I.B.), the Facial Pain Research Foundation (A.I.B.), and US NIH grant P01DA009158 (A.M.). We thank C. Webb for help with an initial CB1R docking screen. We thank B. Ahanou for help analyzing the open field data. We thank Dr. Haoqing Wang (cEMc and Sarafan ChEM-H) for help with cryoEM. We thank M. M. Rachman for editing the manuscript. We gratefully acknowledge OpenEye Software for Omega and related tools and Schrödinger, Inc. for the Maestro package. Select agonist functional data was generously provided by the National Institute of Mental Health's Psychoactive Drug Screening Program, Contract # HHSN-271-2018-00023-C, directed by B. L. Roth.

## Author contributions

T.A.T. and R.M.S. conducted the docking screens with input from B.K.S. Ligand optimization was performed by T.A.T. and C.I.-T. with input from B.K.S., and A.M. N.K.T., S.G., F.T., and Y.L. performed binding or functional assays with input from T.A.T., C.I.-T., and A.M. K.K. prepared the CB1-G$_i$ complex, collected cryo-EM data with help from E.S.O., and modelled the structure with help from F.L. K.K. collected bimane data with help from Y.S. T.A.T. and J.M.B. did the drug formulations for in vivo experiments. J.M.B. performed and analyzed the in vivo pharmacology experiments assisted by T.A.T., V.C., S.R.R., K.B., and J.B., supervised and coanalyzed by A.I.B. E.S.O processed data and obtained the cryo-EM map. Y.S. performed the GTP-turnover assays with help from K.K. and E.S.O. H.K. and C.N. tested select compounds in the panel of G protein and β-arrestin subtypes with supervision from M.S. and L.S. T.A.T., C.I.-T. and T.C.H. helped design bespoke analogs with supervision from A.M. I.G. performed the colloidal aggregation screens. Y.S.M. and D.S.R. supervised compound synthesis of Enamine compounds purchased from the ZINC15 database and 12 billion catalog. J.J.I. built the ZINC15 libraries. B.K.S., A.I.B., A.M., and K.K. supervised the project. T.A.T. wrote the paper with input from all other authors, and primary editing from B.K.S.

## Competing interests

B.K.S. is co-founder of BlueDolphin, LLC, Epiodyne, and Deep Apple Therapeutics, Inc., and serves on the SRB of Genentech, the SABs of Schrodinger LLC and of Vilya Therapeutics. J.J.I. is a co-founder of BlueDolphin LLC and of Deep Apple Therapeutics. Y.S.M. is a VP of Sales and Marketing at Enamine Ltd. and a scientific advisor at Chemspace LLC. D.S.R. is an employee of Enamine, Ltd. H.K., C.N., M.S., and L.S. are employees of Domain Therapeutics North America Inc. The authors declare no other competing interests.
