## [Transparent Peer Review file · Nature Communications]

Virtual library docking for cannabinoid-1 receptor agonists with reduced side effects

Corresponding Author: Professor Brian Shoichet

Version 0:

Reviewer comments:

Reviewer #1

(Remarks to the Author)

The article by Tummino et al describe the use of ultra large docking to discover CB1 agonists with new chemotypes. The article is well-written, and the experiments are detailed. The results are noteworthy. In general, this work will be useful to generate candidates to study CB1 (CB2) and similar GPCRs.

There are some issues that the authors should address.

- 1350 is a pyrazole scaffold so it is not entirely a novel chemotype (Rimo-like) Similar amide -S383 interaction is expected.
- Can the authors comment on the mechanism for 1350 being different than CP55 in tetrad, considering it is expected to adopt a similar shape as MDMA-Fub and possibly CP? 1350 has no bias in the signaling, so in general the analgesia is expected to bring the normal CB1 side-effects. The CB1/CB2 selectivity is moderately different for 1350 and CP55, yet that may not entirely explain the differential behavior. Does 20-fold therapeutic window for analgesia over psychotropic effects translate in higher species? Does 1350 distinguish itself from CP or similar CB1 agonists to be therapeutically useful?
- Can the authors comment on MPO scores of 1350 and similar compounds and in general the filter used for docking?
- Was 1350 tested for oral PK, metabolic stability, PPB etc for 1350?
- 1350/4042 are esters which can get hydrolyzed with acid not retaining CB1 activity in vivo as seen in the carboxylate analog, So the LLE of 4.7 for 1350 may not translate.
- N-methylation in 1066 abolished CB1 binding, Did the authors try N methylation on 1350.
- There is no clear synthetic scheme for the bespoke compounds. Synthetic intermediates have not been mentioned except carboxylic acid, azide etc and ref 67.

Clear synthetic procedures using X intermediate and Y intermediates gave the compound needs to be included to allow for work to be reproduced following the methods in the manuscript.

- Rotation values for enantiomers need to be reported. How was the stereochemistry assigned, is it based on cryoEM data?
- Minor: Page 1 line 31 the font for "analgesia, hypothermia, catalepsy, and 31 hypolocomotion, the latter three of which may be considered adverse drug reactions. Meanwhile, inconclusive results in human clinical trials 12 32 have led to uncertainty" etc is different.
- Define the acronym ECFP4-based molecular fingerprints.
- Figure 1 C. and E. represent mean \pm SEM from three independent experiments. Where they run in duplicate or triplicate?
- In the Discussion section "these new ligands to a 1.9 nM Ki full agonist of the CB1R ('1350') is it 1.9 nM or 0.9 nM

Reviewer #2

(Remarks to the Author)

The submission "Large library docking for cannabinoid-1 receptor agonists 2 with reduced side effects" by Tummino et al describe the development of new agonists for the (CB1R) orthosteric site, for which authors screened 74 million lead-like molecules from ZINC database using DOCK and then prioritized 46 high scorers for experimental testing. The prioritization relied on respective chemical diversity of the hits, their dissimilarity to known CB1R ligands and significant expert knowledge on protein-ligand interactions critical for agonistic action of the receptor.

As the result of experimental evaluation, 9 out of 46 compounds demonstrated >50% ligand displacement in radiolabel assay; the most active hit with $K_i = 0.7 \mu\text{M}$ was further optimized into 'compound 1350' that demonstrated 0.95 nM potency

as a full CB1R agonist. Another advanced compound '4936' demonstrated 7nM effect. Both '1350' and '4936' have also been evaluated in vivo to tackle their off-target effect and PK profiles; based on those, '1350' has been selected as the lead. Notably, the binding of 'compound 1350' to the target was validated by the cryo-EM, with the original docking pose confirmed and validated.

The analgesic properties of '1350' in various pain models have further been investigated and demonstrated a 2-20-fold therapeutic window over hypolocomotion, sedation, and catalepsy and exhibited no conditioned place preference. The reviewers could suggest several minor revision points, such as I am not sure if 74M docking campaign qualifies as 'large-scale' in modern realities, where recent docking studies routinely operate on the scales of 100s or 1000s of millions of molecules. Similarly, the statement that "large library docking can reveal unexpected chemotypes" is not really supported by any analysis involving the size of the docking library.

Rather, the strength of this work seems to be in a synergetic use of methods of docking, qsar, structural biology, significant medicinal chemistry and the use of expert insight that collectively led to remarkable results with very significant practical implications.

This is a solid and impressive work, a stellar example of the synergetic use of the above methodologies. However, some 'take home' messages of this study need to be adjusted, and the success of the work should be attributed to the expert- and synergetic use of computational and experimental approaches.

Reviewer #3

(Remarks to the Author)

Reviewer #4

(Remarks to the Author)

This study focuses on the discovery and characterization of new chemotypes for the Cannabinoid CB1 receptor (CB1R) using structure-based approaches. The authors demonstrate that novel small molecules can be identified through virtual screening of the active structure of CB1R. These molecules exhibit new pharmacological properties with reduced side effects typically associated with cannabinoids in a rodent model. The potential significance of this study lies in its demonstration of how structural information, combined with virtual screening of small molecule, can be utilized to discover new ligands with unique properties that act at the orthostatic site of receptors, as previously also shown by this group and others for other GPCRs. However, while the findings are promising, several issues, questions, and the need for further validation must be addressed for the study to be fully convincing and support some conclusion. Additionally, a better reporting and presentation of data, as well as certain limitations require more thorough considerations. Specific details are provided below:

Specific points:

Abstract: "1350, a 0.95 nM ligand and a full CB1R agonist". It is a full agonist only on Gi/o-mediated responses, but not on G13 and β -arrestin responses. This distinction is important to make. Nowadays, agonistic activity needs to be defined based on the specific response or signaling pathways they engender, as many ligands exhibit biased properties and can act as full agonist on a response/pathway, but not another.

It is unclear what the author means by "the molecules were visually evaluated." It would be beneficial to the future readers if the author could elaborate on this point.

Fig 1F: "Similarly, all four ligands exhibit aromatic stacking and hydrophobic packing with the twin-toggle switch residues W356.48 and F2003.36". From the figure, such conclusion seems to be supported for '51486, and '0450, but less evident in the case of '7800 and '7019 with F2003.36. How was it this conclusion reached? Was any molecular dynamics with RMSD and frequency of interaction of key residues with important pharmacophores in each molecule, performed? Moreover, why if all molecules engage the active twin-toggle switch '7019 seem to behave as antagonists (Fig. S2A and S6) (see also comment below)?

"Given the structural similarities and potency differences, '4051, '1066, and '4388 may be used as inactive "probe pairs" in future research." It is unclear what the authors mean, here. Functional data show that '4388 may act as an inactive probe (antagonist or even inverse agonist, Fig. S6), but '4051 is still an agonist, although with low potency. Functional data with '1066 were not found, but that might be an oversight by this reviewer. Be that as it may, there is a missed opportunity here to validate the antagonistic properties of some molecules and more importantly to provide mechanistic insights into their lack of activity based on structure data/binding poses or even MD simulation.

Fig 2B: The author could highlight where the nine hits are in the figure.

"The leads that emerged, '1350 and '4936 are both potent binders of CB1R, with '1350 at 0.9 nM being 3-fold more potent...." Binding data should at least be done for '1350 on CB2R, to link selectivity and physiological/adverse effects data (or lack thereof). Similarly, binding analysis of the reference ligand CP55,940 on CB2R should be performed and presented

in the study.

The rationale for using redundant signaling and G protein β -arrestin assays throughout the study is unclear to this reviewer and needs clarification and rationalization in the text. For instance, why using Tango β -arrestin-2 vs the DiscoverX PathHunter β -arrestin-2 assay vs bioSensAll for the characterization of small molecule considering differences in sensitivity of these assays? Similarly, why using the TRUPATH vs bioSensAll G protein assays and different cAMP Inhibition Assay. There are clear differences in the pharmacological parameters extracted from these experiments (Suppl. Tables), and it becomes difficult to interpret and compare the pharmacological characteristics of ligands amongst them and between receptor subtype and species, considering the variability in sensitivity of these assays. This needs to be addressed.

It is challenging for this reviewer (and potentially for future readers) to fully appreciate if differences (or lack thereof) in responses between rCB1 and hCB1 exist, due to the variety of assays used for the same pathway/response and the way data are presented throughout the study. I would suggest presenting similar data on binding and signaling of rCB1 and hCB1 in the main figures (as well as in Suppl. Tables comparing rCB1 and hCB1). Additionally, and as underscore in my previous comments, some figures are mislabeled, further complicating the analysis. Are the affinity, potency, and efficacy of ligands consistent between hCB1 and rCB1 across different responses and assays? If not, what do the structural information and binding poses of the ligands in each receptor subtype tell us about these differences and similarities? It is crucial to address this point because understanding whether structure-based approaches used on rodent CB1R are translatable to the human subtype receptor is essential for ultimately improving therapeutics (in human) and validating their approaches. If limitations exist, they need to be discussed.

To improve readability and appreciation of the work, I caution the author to pay close attention to reporting accurately the figures and table in text. In some instances, information is lacking, not complete or erroneous (e.g. mislabeling of figures). I only provide some examples here: "Owing to coupling to the inhibitory G α i G protein, functional efficacy experiments monitoring a decrease in forskolin (FSK) simulated cAMP were tested using hCB1-expressing cells, with '51486 and '0450 showing modest agonist" Add Fig.S2A at the end of this sentence. Figure S4: hCB1 binding and functional data for analogs. However, the labeling of the figure in S4A, says rCB1. Supplementary Table 2: what is 0.026 in Cerep cAMP data for CP-55,940 referring to? What are the significance stars referring to? pEC50? or Emax? I suggest putting significance stars next to each value. The author should explain the CP-55,940 data point in Fig. S4C and S7B, since they report in the Methods that 30 nM of reference ligand was used, but the data are expressed as % of 10 nM CP-55,940. How can this be?

"In summary, '1350 shows no functional selectivity but is both more potent and more relatively efficacious than CP-55,940 at CB1R but not CB2R." '1350 showed reduced Emax but better potency than CP-55,940. It is surprising that it did not show difference in functional selectivity since changes in relative activity is generally outweighed by changes in potency (which is in log scales) over Emax, which span is limited by the nature of the assay/sensor or response.

Fig. S5-S7: In many instances it is unclear how BRET functional data were fitted, as some curves are not even interpolating data sets (Ex: Fig. S5 A, Gz 2-AG and '8690; G13 and G15 for 2-AG; and β -arrestin for hCB1 and '8690, etc.). Why for some BRET sensors, basal responses seem to have been normalized to 0, while for other responses, not. Moreover, it is unclear how EC50 interval between 1618 – 398200 for G13 responses can lead into a pEC50 of 5.4 [5.1 – 5.8] with 95% confidence (Suppl. Table 4). Gz is generally a sensitive sensor and responsive to CB activation, but its response is highly variable for the high affinity ligand CP-55,940 but not for the lower affinity ligand 2-AG, why? While Gs and Gq activity for both CB1R and CB2R has been assessed, there are no positive controls for such response.

Discussion: The authors should exercise caution when concluding on structure-based approaches to identify new CB1 chemotypes with novel properties. Some observed effects may be incidental and not related to the ligands' efficacy, potency, or signaling selectivity. Also, although many identified ligands displayed docked poses that mirrored the interactions of known cannabinoids, some acted as putative antagonists (showing no effect on signaling) or even as inverse agonists (e.g., 1082, 4388, 7019, which this reviewer found interesting). Given that these ligands generally posed similarly on CB1R, it is important to address or discuss why they exhibit distinct functional behaviors, antagonistic and/or inverse agonist ones, and potential bias properties. For instance, was it predictable from the structure of CB1R used (and cryo-EM of the receptor with G α i) and poses of molecules, to predict that some ligands (e.g. 1350) would behave differently in terms of other G proteins and β -arrestin responses (as compared to the reference ligand)? I suspect not.

Discussion and results: "Additionally, in our hands using CB2R knockout mice, at minimum the analgesic effects of '1350 are not due to engagement of CB2Rs" What about the tetrad effects? Analgesic effects were tested in CB1 and CB2 KO mice, but side effects were not tested in these mice.

Methods "The HEK293 clonal cell line (HEK293SL cells) for bioSens-All experiments were derived from HEK293 cells purchased from ATCC." The authors should acknowledge the source and validation of cells in the text, since I believe they did not derive them.

(Remarks to the Author)

This is a comprehensive research paper; the authors demonstrate a full picture of preclinical drug discovery targeting CB1 receptors to develop new chemotypes with better therapeutic windows and fewer adverse effects. Based on the previously experimental structures, the author first conducted large library docking to identify and filter novel hits that retain favourable physical features and the potential to form key polar interactions with CB1R. One series was subsequently expanded and optimized to result in two potent lead compounds '1350 and its derivative '4936. The systematic SAR was built up with the docking poses of lead compounds and validated by the cryo-EM structure of CB1R in complex with '1350. These compounds were well characterized in the pharmacology regarding selectivity (CB1R vs CB2R) and biased agonism, and further investigated in vivo analgetic efficacy and other cannabinoid tetrad of behaviors. This study advances our understanding of the therapeutic profiles of CB1R agonists for pain treatment. However, it is very hard to read the manuscript due to the lack of context information and clear writing flow and I recommend a major revision of the text. Examples:

1. Although '1350 is almost equipotent to CB1R and CB2R in pharmacology assays, the knockout animal revealed its analgetic effect mediated via CB1R but not CB2R. This is an interesting finding, suggesting the different participation of cannabinoid agonists in pain physiology. However, neither the introduction nor the discussion provides any background information about the different roles or profiles of two CB receptors, or the rationale for assessing the selectivity.

2. In line 208-211: "The trifluoromethyl group is complemented by van der Waals and quadrupole interactions with residues W2795.43 and T1973.33, as anticipated by the docked structure, and consistent with the improvement in affinity by -1.7 kcal/mol (17-fold in K_i) on its replacement of the original fluorine."

I assumed the "-1.7kcal/mol" originates from the binding free energy calculation based on docking results, but I did not find any corresponding method description or raw data comparison between '1350 (trifluoromethyl) and '51486 (fluorine).

3. In line 164-167: "51486, the lipophilic ligand efficiency (LLE) of '1350 improved from 3.1 to 4.7 (Fig.2B), whereas '496_'s LLE stayed approximately the same (3.2); both compare favorably to an LLE of 2.6 for the positive control CP-55,940, which is substantially more hydrophobic than either of the two new agonists.

The authors assumed the reader understood the calculation of $LLE = pIC_{50} - \log P$. However, it seems the $\log P$ value of CP-55,940 is not provided in this paper using the consistent method used for the two new leads, but only acclaims the more hydrophobicity of CP-55,940. I suggest authors to describe the concept of LLE in the method or figure legend and include $cLogP$ value of CP-55,940 in the text or figures.

4. In pharmacology assessment, Fig S5, 2-AG was used as positive control? But this information was not mentioned in method or anywhere. Fig S6, the compound labels are not clear, such as '3737 a-d; and '7019-31/29; do this mean different batch of synthesis, please clarify this information.

In addition, cryo-EM structure of '1350 was used to confirm the docking pose. However, in consistent with bad local resolution (Fig.S8B), the density for ECL2 and other regions surrounding the ligand is fragmented and ambiguous, not enough to determine the rotamer of some residues. These undetermined sidechains include H178, S383, S173 and the former two are key interactions that the authors acclaimed in the text and figure 3. After looking through the method, I strongly suggest that authors are supposed to improve the EM map quality as they gained a reasonable number of good particles. Here are several usual ways may help.

1. Try polishing the final particle stacks on relion/5.0 and import to cryosparc for final 3D refinement.

2. Try local refinement in cryosparc or relion using masks only cover the receptor region or the receptor-Gi region; the similar approach was used in CB1 receptor structure(8GHV).

3. Try the latest relion/5.0 of 3D refinement with Blush regularisation.

I fully understand that maybe the time would not allow authors to optimise the map quality. In such a case, I suggested authors mention the resolution limitation and rephrase some words more cautiously. For example,

In line 206-208, "The major interactions with CB1R predicted by the docking are preserved in the experimental structure, including the key hydrogen-bond between the amide carbonyl of the ligand and S3837.39" maybe rephrase to "despite the local resolution limit that prevents unambiguously modelling all rotamers,are likely to be preserved...."

Furthermore, the density display in Fig.3A is ambiguous at too low counter level. Please referred to the Fig S2D in the published CB1R structure (6N4B, [https://www.cell.com/cell/fulltext/S0092-8674\(18\)31565-4?_returnURL=https%3A/linkinghub.elsevier.com/retrieve/pii/S0092867418315654%3Fshowall=true](https://www.cell.com/cell/fulltext/S0092-8674(18)31565-4?_returnURL=https%3A/linkinghub.elsevier.com/retrieve/pii/S0092867418315654%3Fshowall=true)) to present the ligand density precisely.

In addition, the ligand of 6N4B (MDMB-Fubinaca) was used to set up the molecular docking, and MDMB-Fub also has similar methyl ester moiety and binding pose to "1350. 6N4B is also used as the initial template for modelling '1350 structure, so I suggested author make a supplementary figure to compare the '1350 structure and the published 6N4B.

Other minor corrections:

1. In line 176: "The one exception is '4936, whose E_{max} of 65% is more consistent with strong partial agonism"

I reckon it's a typo for the compound '4936

2. The figure legend for S4: panel A. "Competition binding data for primary hits and a subset of their analogs at hCB1." This is the binding assay using rat brain so it should be rCB1 not hCB1.

Version 1:

Reviewer comments:

Reviewer #1

(Remarks to the Author)

The authors have satisfactorily commented on the points raised and included additional data in many cases. However, the authors' response to the below are not satisfactory:

1.7

The synthesis section has been improved and yields reported, yet it is not clear on what scale each reaction leading to the product was run giving the respective product yields? Are these run on a 100 mg scale for each reaction shown in general methods section or the mmol/mol different in each case? Even if it is run on a 100 mg scale the mmol values for each reactant will be different. This is generally required to understand for the scalability of the methods vis-a-vis the reported yields.

1.8 Rotation values for enantiomers need to be reported. How was the stereochemistry assigned, is it based on cryoEM data? We do not have rotation values for these compounds.

The authors' response is: The polarimetry experiments are expensive to run and would require a minimum of 50 mg of each enantiomer (and up to 500 mg, depending on the cell size), which would cost at least \$10,000 of compound material. We beg the Reviewer's indulgence here. That said, we have been able to assign stereochemistry based on the cryoEM data, as only one enantiomer fits the density.

In general, the sign of rotation and values could be easily obtained with 1-2 mg of compound. Many academic labs or CROs (Enamine?) should have accessible polarimeters that may be able to get this data. The cost and amount of compound estimated for such is overestimated. Hence the authors should make every effort to get the data. The S/R enantiomer without the sign of rotation and values is incomplete.

Reviewer #2

(Remarks to the Author)

I am satisfied with the revisions. My criticism points were sufficiently addressed.

Reviewer #3

(Remarks to the Author)

Reviewer #4

(Remarks to the Author)

This is a revision of a previously reviewed study. The authors have generally addressed our prior comments and concerns well, incorporating new data and making appropriate revisions to the text. However, a few points still warrant further consideration.

4.10: We apologize for not clearly conveying our thoughts on functional selectivity (bias). Our intention was to emphasize caution when calculating and interpreting bias using E_{max}/EC_{50} values and relative activity, as these parameters are not equally weighted, as EC_{50} influence the relative activity more than E_{max} which is limited in span because of the assay. While this may not constitute statistical bias, compound 1350 shows a clear reduction in E_{max} for G13 and, to some extent β -arrestin, which could have some physiological importance. Our comment was more editorial, as this is the standard way biases are reported in the literature, which we are also bound to use in our many studies, but clearly imperfect and can lead to misinterpretation by the reader.

4.11: We regret any confusion regarding our comment on the lack of positive controls for Gs and Gq signaling. Our concern was not about testing other ligands for CB pathways showing coupling to Gs and Gq, but rather the absence of controls validating the assay with known Gs and Gq receptors.

Reviewer #5

(Remarks to the Author)

Very glad to see the density improvement for the ligand and interactive residues in the new local refinement cryo-EM maps. Well done! Current PDB validation report shows only the focus map and receptor regional model has been deposited. Please deposit both the local refinement map (receptor-focused, as additional map) and the previous consensus CB1-Gi complex map into your PDB deposition due to the molecular integrity. Based on the same principle, please merge the new receptor model into your complex model and deposit the intact complex.

Except from the above small comment on structural data deposition, current version addressed all my concerns and I'm also pleased to accept this manuscript to publish.

PS the last tiny thing to fix is in the Fig 2D, where the molecular profile of 4936 was wrongly pasted.

Version 2:

Reviewer comments:

Reviewer #1

(Remarks to the Author)

The authors have incorporated requested revisions.

Reviewer #3

(Remarks to the Author)

Reviewer #4

(Remarks to the Author)

This is a re-revision of the manuscript. The authors have diligently addressed all my comments.

Reviewer 1

1.1. *1350 is a pyrazole scaffold so it is not entirely a novel chemotype (Rimo-like) Similar amide -S383 interaction is expected.*

This is well-taken. We have added a qualifier on this point to caveats paragraph of the Discussion (p. 26) on novelty. We did already mention that an interaction with Ser383 is ubiquitous among CB1-ligand complexes (p.8, "Similarly, all four are predicted to hydrogen-bond with S383⁷⁻³⁹, a potency-determining interaction at CB1 receptors observed in nearly all agonist-bound ligand-receptor complexes^{48,49}."). We now write, (p.26)

"First, while our lead molecules are dissimilar to known cannabinoids by molecular fingerprint, they do share physical features with many of the characterized ligands, and even some core chemotypes. For example, antagonists containing pyrazole-amides (e.g., rimonabant), which forms part of our most active series, are well-known."

1.2a. *Can the authors comment on the mechanism for 1350 being different than CP55 in tetrad, considering it is expected to adopt a similar shape as MDMB-Fub and possibly CP? 1350 has no bias in the signaling, so in general the analgesia is expected to bring the normal CB1 side-effects. The CB1/CB2 selectivity is moderately different for 1350 and CP55, yet that may not entirely explain the differential behavior.*

We understand where the Reviewer is coming from. Accordingly, we discuss this problem at some length in the Discussion (page 26, reproduced below), and have added further speculation given the new CB2R knockout data.

"Several caveats bear mentioning. First, while our lead molecules are dissimilar to known cannabinoids by molecular fingerprint, they do share physical features with many of the characterized ligands, and even some core chemotypes. For example, antagonists containing pyrazole-amides (e.g., rimonabant), which forms part of our most active series, are well-known. Second, the mechanistic bases for the disentanglement of sedation and catalepsy from analgesia remains uncertain. Often, clear differences in functional selectivity (signaling bias) or subtype selectivity explain the phenotypic differences among ligands^{34,35,55,65}. Here, functional-selectivity differences between '1350, which features two reduced "tetrad" behaviors, and CP-55,940, which does not, were modest, with the only notable difference being the recruitment of G₁₃; the *in vivo* effect of such a finding is not currently understood. Pronounced differences were, however, seen in the functional effects of '1350 on the CB1R and CB2R subtypes. Surprisingly, we observed an increase in catalepsy but no change in analgesia of '1350 in the CB2 knockout mice (**Fig. 11C-D**). However, the mechanism underlying how partial agonism of CB2 would decrease cataleptic behaviors requires further validation. Taken together, we suspect that the separation of analgesic and "tetrad" behaviors in the new molecules may reflect a combination of pharmacokinetic, pharmacodynamic, selectivity, and signaling, though without further investigation this remains speculative. For now, we can only lay the ability to disentangle analgesic efficacy from "tetrad" adverse reactions at the door of the new chemotypes explored⁶⁶⁻⁶⁸. Whether the increased therapeutic window seen here in mice translates to higher-order species also remains to be explored. Finally, we note that while only agonists emerged from the optimization of the initial docking actives, these early docking hits spanned a wide range of chemotypes, and in early assays did not show strong agonism; we cannot rule out that some of them were antagonists, even though only agonists were sought. Docking, in our hands, remains better at finding ligands than making functional distinctions between them, such as predicting agonist or antagonist effects."

1.2b. *Does 20-fold therapeutic window for analgesia over psychotropic effects translate in higher species? Does 1350 distinguish itself from CP or similar CB1 agonists to be therapeutically useful?*

Whereas ‘1350 **does** distinguish itself from CP-55,940 and other cannabinoids by increased therapeutic window, we regret that we have not tested the molecules in higher species. For a manuscript that is already freighted with many experiments, we feel that doing so would be beyond scope, and beg the Reviewer’s and Editor’s indulgence on this point. We do add a caveat to this effect in the Discussion (page 26), writing, “Whether the increased therapeutic window seen here in mice translates to higher-order species also remains to be explored.”

1.3 *Can the authors comment on MPO scores of 1350 and similar compounds and in general the filter used for docking?*

We now provide a **Supplementary Table 13** with MPO values for ‘1350, ‘4936, CP-55,940, WIN 55,212,2, MDMB-Fub, and Rimonabant. All these molecules fall between 3-4 moderate MPO scores, and our new compounds are in the middle of the distribution- ‘**1350** = 3.7, ‘**4936** = 3.3, CP-55,940 = 3.0, WIN 55, 212,2 = 3.7, MDMB-Fub = 4.3, Rimonabant = 3.1. We have added text to page 16 and the methods section to this effect.

Apropos of docking, MPO values are not explicitly used as filters. However, we do filter for molecular weight (350-500 Da) and cLogP (3 – 5) in the docking, as well as for pKa ranges (here, we only docked molecules calculated to be uncharged at pH 7.4). We chose these parameters after a trial docking campaign with more polar and smaller chemical space found no hits. After the docking, we filtered for structural diversity by clustering the top 300,000 molecules using ECFP4 TC of 0.5, chemical novelty (ECFP4 TC < 0.36 from a list of known CB1 and CB2 ligands with activities < 10 uM from ChEMBL). Distance filters were used to identify molecules that are within H-bond distance to SER-383 with an optional filter for potential to H-bond to HIS-178. All of these are described in the docking methods.

1.4. *Was 1350 tested for oral PK, metabolic stability, PPB etc for 1350?.*

We didn’t test ‘1350 for oral PK, as the compound was only dosed intraperitoneally. We have since tested ‘**1350** for mouse plasma protein binding, MDCK-MDR permeability, aqueous solubility and mouse plasma stability (new **Supplementary Tables 9 – 12** and **Figure S10E-I**). We have added text describing these results (p.16):

“To further explore ‘**1350**, we tested it for aqueous solubility (**Supplementary Table 9**), mouse plasma protein binding (**Supplementary Table 10**), mouse plasma and microsomal stability (**Fig. S10E&F, Supplementary Tables 11 & 12**), membrane permeability, and P-glycoprotein (P-gp) activity (**Fig. S10G-I**; see **Methods**). The molecule was soluble to 23 μM in PBS and was 94% plasma protein bound, values that are perhaps respectable for a lipid receptor ligand. Greater liabilities were seen in its relatively low stability in plasma (41-minute half-life) and relatively high lability in liver microsomes (**Supplementary Table 12**). Conversely, ‘**1350** was relatively membrane-permeable (**Fig. S10G**) and was not a substrate of P-gp (**Fig. S10H**). These observations are broadly in line with its physicochemical properties (cLogP = 4.08, cLogD = 4.50) and *in vivo* pharmacokinetics, where its plasma half-life is 111 min and where its level in the CSF is below the quantification limit (speaking to its low free fraction in the brain). Still, CP-55,940 is efficacious *in vivo* despite similar ($T_{1/2, \text{brain}} = 127$ min) or in some cases worse (cLogP = 5.66, cLogD = 5.90, $C_{\text{max}} = 19.2$ ng/g versus 44.1 ng/g) physicochemical and pharmacokinetic properties. Further, the metabolic and pharmacokinetic profiles for ‘**1350**, ‘**4936**, and CP-55,940 are in line with their moderate druglike central nervous system multiparameter optimization (MPO)⁵⁷ values, which are higher (3.3 and 3.7 for ‘**1350** and ‘**4936**, respectively versus 3.0 for CP-55,940) for the novel ligands (**Supplementary Table 13**), suggestive of moderate alignment with key CNS drug desirability properties. Taken together, ‘**1350** is not only the most potent ligand we discovered, but it has a modestly favorable pharmacokinetic profile and is therefore the focus of the proceeding efficacy experiments.”

1.5 *1350/4042 are esters which can get hydrolyzed with acid not retaining CB1 activity in vivo as seen in the carboxylate analog, So the LLE of 4.7 for 1350 may not translate.*

Conversion to the carboxylate decreases activity of this series >5000-fold (e.g., compound '4051), so we think the LLE of 4.7 reflects the activity, both in vitro and in vivo, of the parent molecule. The problem is the reduction of half-life to which the ester contributes. That said, several other cannabinoids have an ester, so it doesn't seem to be entirely damning.

1.6 *N-methylation in 1066 abolished CB1 binding, Did the authors try N methylation on 1350.*

Given the results with 1066, we did not try this change as we expected it would kill activity here, too. We have added text to page 11 to this point:

"Though we did not try combining methylation of both the amide nitrogen and the chiral center (addition of N-Me to '1350, '4936, or '5806), we expect this too would negatively impact binding."

1.7 *There is no clear synthetic scheme for the bespoke compounds. Synthetic intermediates have not been mentioned except carboxylic acid, azide etc and ref 67. Clear synthetic procedures using X intermediate and Y intermediates gave the compound needs to be included to allow for work to be reproduced following the methods in the manuscript.*

We regret this oversight. We have updated the Supplemental information file with more detail on the synthetic schemes, including specific starting materials and their catalog numbers for each compound.

1.8 *Rotation values for enantiomers need to be reported. How was the stereochemistry assigned, is it based on cryoEM data?*

We do not have rotation values for these compounds. The polarimetry experiments are expensive to run and would require a minimum of 50 mg of each enantiomer (and up to 500 mg, depending on the cell size), which would cost at least \$10,000 of compound material. We beg the Reviewer's indulgence here. That said, we have been able to assign stereochemistry based on the cryoEM data, as only one enantiomer fits the density.

1.9. *Minor: Page 1 line 31 the font for "analgesia, hypothermia, catalepsy, and 31 hypolocomotion, the latter three of which may be considered adverse drug reactions. Meanwhile, inconclusive results in human clinical trials 12 32 have led to uncertainty" etc is different.*

Fixed, thank you.

1.10 *Define the acronym ECFP4-based molecular fingerprints.*

We have added this to page 5 including a reference to the ECFP method (Rogers & Hahn, Journal of Chemical Information and Modeling, 2010)

1.11 *Figure 1 C. and E. represent mean \pm SEM from three independent experiments. Where they run in duplicate or triplicate?*

They were run in triplicate, yes. Text was added to Figure legend 1 to this point.

1.12 *In the Discussion section these new ligands to a 1.9 nM Ki full agonist of the CB1R ("1350") is it 1.9 nM or 0.9 nM*

Thank you for catching this; it has been fixed to 0.95 nM Ki and full agonist.

Reviewer 2

The reviewers could suggest several minor revision points, such as I am not sure if 74M docking campaign qualifies as 'large-scale' in modern realities, where recent docking studies routinely operate on the scales of 100s or 1000s of millions of molecules. Similarly, the statement that "large library docking can reveal unexpected chemotypes" is not really supported by any analysis involving the size of the docking library.

Rather, the strength of this work seems to be in a synergetic use of methods of docking, qsar, structural biology, significant medicinal chemistry and the use of expert insight that collectively led to remarkable results with very significant practical implications.

This is a solid and impressive work, a stellar example of the synergetic use of the above methodologies. However, some 'take home' messages of this study need to be adjusted, and the success of the work should be attributed to the expert- and synergetic use of computational and experimental approaches.

2.1 *I am not sure if 74M docking campaign qualifies as 'large-scale' in modern realities, where recent docking studies routinely operate on the scales of 100s or 1000s of millions of molecules.*

This is well-taken. We have modified the title of the paper to "Virtual library docking for cannabinoid-1 receptor agonists with reduced side effects" and removed the word "large" throughout. We would note that a 74 million molecule library is large by the previous standards of the field, as recently as 2019, and even today many docking campaigns screen fewer molecules. Still, we agree that this is not the level of the more recent 1 to 10 billion molecule screens described by Katritch and colleagues, Arthanari and colleagues, and by us.

2.2 *Similarly, the statement that "large library docking can reveal unexpected chemotypes" is not really supported by any analysis involving the size of the docking library.*

As above, we have removed "large" from this sentence. In the introduction, we cite ten examples (from our lab, and others, refs. 29-39, page 4) where large libraries were employed to find novel chemotypes, including a recent paper on the specific analysis of the impact that virtual library size has on virtual screening (ref 29, page 4). With respect, we do feel that one of the things docking has consistently delivered over the last decade is compound novelty. Recent studies suggest that this, in turn, is improved by library size (Lyu et al., *Nature Chem Biol*, 2023; Liu et al., S).

2.3. *Rather, the strength of this work seems to be in a synergetic use of methods of docking, qsar, structural biology, significant medicinal chemistry and the use of expert insight that collectively led to remarkable results with very significant practical implications. This is a solid and impressive work, a stellar example of the synergetic use of the above methodologies. However, some 'take home' messages of this study need to be adjusted, and the success of the work should be attributed to the expert- and synergetic use of computational and experimental approaches.*

We have adjusted the Discussion to emphasize the impact of these different technologies (p. 27).

"Despite these caveats, the main observations of this study seem clear. Employing synergistic computational and experimental approaches, including molecular docking, computational chemistry, medicinal chemistry, structural biology, in-depth molecular and *in vivo* pharmacology, new cannabinoid-1 receptor analgesics with reduced *in vivo* side effects were discovered. A cryo-EM structure of the '1350-CB1-G_{i1} complex confirmed its docking-predicted pose. The new agonists are analgesic in several of behavioral assays, and unlike the control cannabinoid CP-55,940 have a 2-20-fold therapeutic window over hypolocomotion, sedation, and catalepsy. We suspect that newer chemotypes still remain to be discovered, and that these might further separate the dose-limiting side-effect aspects of the cannabinoid tetrad while maintaining analgesic potency, supporting the development of new cannabinoid medicines to treat pain."

Reviewers 3/4

4.1. Abstract: “1350, a 0.95 nM ligand and a full CB1R agonist”. *It is a full agonist only on Gi/o-mediated responses, but not on G13 and β -arrestin responses. This distinction is important to make. Nowadays, agonistic activity needs to be defined based on the specific response or signaling pathways they engender...*

We have added “of G_{i/o} signaling” to the abstract and throughout the text to clarify this.

4.2. *It is unclear what the author means by “the molecules were visually evaluated.” It would be beneficial to the future readers if the author could elaborate on this point.*

Visual examination has been standard in our workflow, and those of others, since the inception of the field (explained in detail in ref. 46, Bender et al., *Nature Protocols* 2021). We agree that this won't be obvious to non-experts and have added text with examples of what is being evaluated (p.5).

“The top-ranking 10,000 remaining molecules were visually evaluated⁴⁶ in UCSF Chimera⁴⁷ for features that are not included or are approximated in the scoring function and chemoinformatic filters, such as angles and distances of hydrogen bonds, dihedral strain, and incorrect protomer or tautomerization.”

4.3. Fig 1F: *“Similarly, all four ligands exhibit aromatic stacking and hydrophobic packing with the twin-toggle switch residues W356⁴⁸ and F200^{3,36}”. From the figure, such conclusion seems to be supported for ‘51486, and ‘0450, but less evident in the case of ‘7800 and ‘7019 with F200^{3,36}. How was it this conclusion reached?*

We agree, we overstated this. We now write:

“The two most potent hits (**‘51486** and **‘0450**) further exhibit aromatic stacking and hydrophobic packing with the twin-toggle switch residues W356^{6,48} and F200^{3,36} which are important for receptor activation^{43,44} and may explain their stronger agonism profiles compared to **‘7019** and **‘7800**, though on-target potency may also play a role.”

Was any molecular dynamics with RMSD and frequency of interaction of key residues with important pharmacophores in each molecule, performed? Moreover, why if all molecules engage the active twin-toggle switch ‘7019 seem to behave as antagonists (Fig. S2A and S6) (see also comment below)?

Molecular dynamics was not used. We did not test the molecules for antagonism, at least partly because the initial hits, though decently potent by the standards of the field, were close to their solubility limits at their apparent EC₅₀ values. This made detailed functional assessment difficult until potency could be improved. Those molecules that were optimized for potency did turn out to be agonists. For the rest of the initial chemotypes discovered, we must suspend judgment as to whether they are really agonists or antagonists—we are confident that they are ligands.

4.4 “Given the structural similarities and potency differences, ‘4051, ‘1066, and ‘4388 may be used as inactive “probe pairs” in future research.” *It is unclear what the authors mean here.*

Following the definitions in the field, probe pairs are structurally-related molecules one of which is active and the other inactive (or much less active). These molecules are the inactive part of a probe *pair* because they are structurally related to our active “probes” (**‘1350** and **‘4936**, 1-2 atom changes per molecule between the pairs) but are 200-5000 fold less potent than the parent probe. Such molecules, used in counterpoint to the actives, can disentangle on- from off-target effects of the probe in cellular or in vivo studies. For instance, in a phenotypic study if the **active** member of the probe-pair confers a phenotype that the **inactive** member does not, one can infer that this phenotype is mediated by CB1R since, given their close structural similarity, the active and inactive molecules will share most of the same off-targets. Conversely, if both are active in the phenotypic assay, the activity is unlikely via the

CB1 receptor. This logic is well-established in the Chemical Biology field. We have included references to two recent papers on this topic to clarify this for the reader.

Functional data show that '4388 may act as an inactive probe (antagonist or even inverse agonist, Fig. S6), but '4051 is still an agonist, although with low potency. Functional data with '1066 were not found, but that might be an oversight by this reviewer. Be that as it may, there is a missed opportunity here to validate the antagonistic properties of some molecules and more importantly to provide mechanistic insights into their lack of activity based on structure data/binding poses or even MD simulation.

For us, the functional differences, to the extent that they are reliable, are not as important as the 3-log order drop in potency. While we agree that a mechanistic understanding of what separates a potent agonist from an effectively inactive molecule, despite the 1-2-atom changes between them, is interesting, we feel that such studies are out-of-scope for a manuscript already freighted with many experiments. We beg the Reviewer's and Editor's indulgence here.

4.5. Fig 2B: *The author could highlight where the nine hits are in the figure.*

We regret that we were not as clear as we should have been, but note that the nine hits from the docking screen can be found in Figure **1D**, not Figure **2B**. Figure 2B shows the singular best hit '**51486** and how it was optimized to the in vivo lead molecules.

4.6. "The leads that emerged, '1350 and '4936 are both potent binders of CB1R, with '1350 at 0.9 nM being 3-fold more potent....." *Binding data should at least be done for '1350 on CB2R, to link selectivity and physiological/adverse effects data (or lack thereof). Similarly, binding analysis of the reference ligand CP55,940 on CB2R should be performed and presented in the study.*

We now provide CB1 and CB2 binding data for CP-55,940, '**1350** and '**4936** in **FigS7A-C**.

4.7. *The rationale for using redundant signaling and G protein β -arrestin assays throughout the study is unclear to this reviewer and needs clarification and rationalization in the text. For instance, why using Tango β -arrestin-2 vs the DiscoverX PathHunter β -arrestin-2 assay vs bioSensAll for the characterization of small molecule considering differences in sensitivity of these assay? Similarly, why using the TRUPATH vs bioSensAll G protein assays and different cAMP Inhibition Assay. There are clear differences in the pharmacological parameters extracted from these experiments (Suppl. Tables), and it becomes difficult to interpreted and compare the pharmacological characteristics of ligands amongst them and between receptor subtype and species, considering the variability in sensitivity of these assays. This needs to be addressed.*

It is true that several signaling assays are explored here, but for good reason. **First**, in our experience it is important to show that novel ligands, especially ones as hydrophobic as these, are active across a range of assays, and are not, for instance, interfering with one specific assay (basically, a control for artifactual activity). **Second**, while we can qualitatively compare activity in different G-protein and arrestin assays, we cannot make quantitative judgements about ligand "bias" using such methods due to system bias, which led us to the comprehensive analysis done with the ebBRET bioSens-All experiments. Such bias has in the past explained in vivo side effect differences between molecules, though it did not do so here. It still seemed worth exploring.

We have included a new sentence on p. 12 toward point 1:

"To verify that the activity is reproducible, we investigated it further in orthogonal G protein, β -arrestin-2, and off-target assays (**Fig. S3-S7, Supplementary Tables 2-7**)."

We have included additional clarifications regarding point 2 on p. 13 including three references on the topic:

“Fortified by this potent activity, and to control for system bias,^{55–57} we investigated ‘1350 for differential recruitment of several G proteins and β -arrestin-2 (“signaling bias”) against both CB1R and CB2R in the ebBRET bioSens-All® platform.”

4.8. *It is challenging for this reviewer (and potentially for future readers) to fully appreciate if differences (or lack thereof) in responses between rCB1 and hCB1 exist, due to the variety of assays used for the same pathway/response and the way data are presented throughout the study. I would suggest presenting similar data on binding and signaling of rCB1 and hCB1 in the main figures (as well as in Suppl. Tables comparing rCB1 and hCB1). Additionally, and as underscore in my previous comments, some figures are mislabeled, further complicating the analysis. Are the affinity, potency, and efficacy of ligands consistent between hCB1 and rCB1 across different responses and assays? If not, what do the structural information and binding poses of the ligands in each receptor subtype tell us about these differences and similarities? It is crucial to address this point because understanding whether structure-based approaches used on rodent CB1R are translatable to the human subtype receptor is essential for ultimately improving therapeutics (in human) and validating their approaches. If limitations exist, they need to be discussed.*

We regret figure mislabeling, and hope we have corrected most or all of those.

Apropos of rodent/human receptor differences, we should clarify that we did not use rodent CB1R for any “structure-based approaches”, but rather docked against the human receptor as defined in PDB 5XR8, and determined the structure of the human CB1R/’1350 complex. We now make this clearer in the text. Regarding the similarities of rCB1 and hCB1, the sequence identity between human and mouse CB1 receptors is >97%. The different residues in rCB1 versus hCB1 are largely in the N- or C-termini, in ECL2 pointing away from the binding pocket, or are at the top of TM2 and TM3 (shown below in red if they are resolved in the structure). I175V and R186P are both pointing away from the binding pocket (ligand shown in blue), toward solvent or membrane, and are unlikely to affect ligand-receptor interactions.

We agree that it would be nice to have full functional **and** binding data against **both** rodent and human receptors. However, that would demand much new work and wouldn't affect the conclusions of this study. Especially for a manuscript already heavy with experiments, we think it out of scope. We used binding affinities against the rodent receptor as estimates to guide ligand design and to prioritize which molecules should be studied in time/labor-intensive signaling studies, which focused on hCB1 due to the desire to ultimately translate to humans. By knock-out, we do show that rodent CB1 is necessary for *in vivo* activity, broadly supporting the idea that hCB1 functional activity translate to the mouse and align with the rodent CB1 binding affinities.

To meet the Reviewer part way, we were able to add comparisons of CP-55,940, '1350, and '4936 at rCB1 and hCB1 in the binding assay in **Fig. S7A-C**.

4.9. *To improve readability and appreciation of the work, I caution the author to pay close attention to reporting accurately the figures and table in text...e.g.:“Owing to coupling to the inhibitory Gai G protein, functional efficacy experiments monitoring a decrease in forskolin (FSK) simulated cAMP were tested using hCB1-expressing cells, with '51486 and '0450 showing modest agonist” Add Fig.S2A at the end of this sentence.*

We have added this additional reference, agreeing that it improves clarity. We have looked for other places to make related changes.

Figure S4: hCB1 binding and functional data for analogs. However, the labeling of the figure in S4A, says rCB1.

We regret this error, which we have fixed, with thanks.

Supplementary Table 2: what is 0.026 in Cerep cAMP data for CP-55,940 referring to?

We agree that this is confusing and have simply removed this point from the figure and tables. For the Reviewer's interest, 0.026 nM is the concentration of CP,55940 that elicits 50% response (EC_{50}) in this assay. It was reported from the CRO that did this particular assay (Eurofins; most other assays in this manuscript were performed by the authors) as a single-concentration data point. Understanding that this can be confusing, we have simply removed the datapoint from the figure and tables—this point is not relevant to the conclusions of this study.

What are the significance stars referring to? pEC50? or Emax? I suggest putting significance stars next to each value.

We regret that the significance stars were not properly labeled in the Supplementary Table. The significance stars were referring to pKi and pEC₅₀ values. However, they were duplicate information from Figure 2G, where the values have already been labeled. For clarity's sake, we have removed them in the Supplementary Table in favor of the main Figure 2.

The author should explain the CP-55,940 data point in Fig. S4C and S7B, since they report in the Methods that 30 nM of reference ligand was used, but the data are expressed as % of 10 nM CP-55,940. How can this be?

The data come from the Eurofins-based experiments, pre-normalized to the effect of CP-55,940 at 10 nM, which is a saturating concentration of CP-55,940 in their hands, rather than to the Emax best fit value of CP-55,940. The 30 nM reference ligand is a mistake and has been removed from the text. We have updated the Methods to clarify these points.

“The results are expressed as a percent of the control response to a saturating concentration of CP-55,940, in this case the 10 nM datapoints. The EC_{50} for the control CP-55,940 was 0.026 nM in the hCB1 assay and 0.082 nM in the hCB2 assay (data not shown).”

4.10. “In summary, '1350 shows no functional selectivity but is both more potent and more relatively efficacious than CP-55,940 at CB1R but not CB2R.” ‘1350 showed reduced Emax but better potency than CP-55,940. It is surprising that it did not show difference in functional selectivity since changes in relative activity is generally outweighed by changes in potency (which is in log scales) over Emax, which span is limited by the nature of the assay/sensor or response.

We should have been clearer. In discussing “functional selectivity” we are talking about “ligand bias”, rather than agonism versus antagonism (the two terms are used interchangeably in the field). The relative efficacy (relative efficacy = $10^{\Delta \log(E_{max}/EC_{50})}$) is a specific term we used to discuss ligands activities (encompassing both their potency and E_{max} values) relative to one another, according to the IUPHAR guidelines for GPCR ligand bias (Kolb, P., Kenakin, T., Alexander, S. P., Bermudez, M., Bohn, L. M., Breinholt, C. S., ... & Gloriam, D. E. (2022). Community guidelines for GPCR ligand bias: IUPHAR review 32. *British journal of pharmacology*, 179(14), 3651-3674.) SI Tables 5 and 7 show the values that were calculated, and Fig S5E and G show the patterns of relative activity at CB1 and CB2. While there were relative efficacy differences for our ligand vs CP-55,940 at some of the effectors, the *pattern* of effectors that was recruited was the same, therefore there is no bias for one pathway over the others. We have added the word “bias” to page 13 to clarify this point.

4.11. Fig. S5-S7: *In many instances it is unclear how BRET functional data were fitted, as some curves are not even interpolating data sets (Ex: Fig. S5 A, Gz 2-AG and ‘8690; G13 and G15 for 2-AG; and β-arrestin for hCB1 and ‘8690, etc.). Why for some BRET sensors, basal responses seem to have been normalized to 0, while for other responses, not.*

In the Methods (p.52) we mention that “If CP-55,940 had no response, data were left unnormalized and uBRET was used for plotting.” So it is not that some BRET sensors are normalized to 0 and some to 100. Instead, we do not normalize the data at all if there was no control response, instead only reporting raw values, these were not included in further analyses. We have added new text into the figure legend to clarify this point, writing:

“If CP-55,940 had no response, data were left unnormalized and raw uBRET was plotted and the values were not included in follow-up analyses. If CP-55,940 had a response, data was normalized to 100% of CP-55,940 activity. The normalized versus unnormalized data is separated by subpanel.”

Moreover, it is unclear how EC₅₀ interval between 1618 – 398200 for G13 responses can lead into a pEC₅₀ of 5.4 [5.1 – 5.8] with 95% confidence (Suppl. Table 4).

This is likely because the curve is missing the top part of the plateau, and curve fitting tries to force a fit even without the top of the curve. We should have not reported best fit values for this curve and we regret this oversight. As 2-AG is not an important molecule for our conclusions, we have decided to remove it from the current draft entirely so as to remove potential confusion.

Gz is generally a sensitive sensor and responsive to CB activation, but its response is highly variable for the high affinity ligand CP-55,940 but not for the lower affinity ligand 2-AG, why?

As mentioned above, we have removed 2-AG from the paper as we are not as confident in its results and believe that they distract from the important comparisons between CP-55940 and ‘1350 or ‘4042. We thank the Reviewer for drawing our attention to this. We suspect that some of these difficulties arise from the challenging physical chemistry of molecules like 2-AG.

While Gs and Gq activity for both CB1R and CB2R has been assessed, there are no positive controls for such response.

Positive controls were not assessed for this assay because largely CB1 is considered a Gi/o receptor. The only potential Gs recruiting CB1 ligand that we know of is HU-210 which is a Schedule I ligand, putting it out of reach for us. To our knowledge, there are no other known CB1 ligands that recruit this signaling partner. Although there are reports of WIN55,212-2 increasing intracellular calcium via recruiting Gq/11 (Lauckner, J. E., Hille, B., & Mackie, K. (2005). PNAS.), this was previously assessed

upon development of this system (Avet... & Bouvier, M. (2022).. *Elife*. and Hauser, ...& & Gloriam, D. E. (2022). *Elife*), and they were not able to see a strong effect of WIN55,212-2 on recruitment of Gq/11. Largely CB1 is considered a Gi/o coupled receptor so it is not a surprise that the control CP-55,940 had no effect in Gq/11 recruitment assays. This is also why for the larger set of molecules that were assessed (Fig S6) we did not look at Gs or Gq/11 signaling at all. We also did not include these values in our radial plots (Fig S5) for the same reason. Finally, there being no control response is why the raw data was provided as described above.

4.12. Discussion: *The authors should exercise caution when concluding on structure-based approaches to identify new CB1 chemotypes with novel properties. Some observed effects may be incidental and not related to the ligands' efficacy, potency, or signaling selectivity. Also, although many identified ligands displayed docked poses that mirrored the interactions of known cannabinoids, some acted as putative antagonists (showing no effect on signaling) or even as inverse agonists (e.g., 1082, 4388, 7019, which this reviewer found interesting). Given that these ligands generally posed similarly on CB1R, it is important to address or discuss why they exhibit distinct functional behaviors, antagonistic and/or inverse agonist ones, and potential bias properties. For instance, was it predictable from the structure of CB1R used (and cryo-EM of the receptor with Gi1) and poses of molecules, to predict that some ligands (e.g. 1350) would behave differently in terms of other G proteins and β -arrestin responses (as compared to the reference ligand)? I suspect not.*

To the Reviewer's point that the observed activities of the new ligands might come from features outside of their signaling, In the Discussion, we now write:

"We suspect that the separation of analgesic and "tetrad" behaviors in the new molecules may reflect a combination of pharmacokinetic, pharmacodynamic, selectivity, and signaling, though without further investigation this remains speculative. ... "

Regarding the challenges of predicting functional activity, we now write, a little further on:

"Finally, we note that while only agonists emerged from the optimization of the initial docking actives, these early docking hits spanned a wide range of chemotypes, and in early assays did not show strong agonism; we cannot rule out that some of them were antagonists, even though only agonists were sought. Docking, in our hands, remains better at finding ligands than making functional distinctions between them, such as predicting agonist or antagonist effects."

As an aside, we did not test the early docking hits for antagonism, at least partly because their activity was only micromolar and so not much better than their solubility limit (a function of the CB1R site). While we share the Reviewer's interests in whether a mix of agonists and antagonists were found—something that we would find attractive, since a mission of docking is to find a range of ligand activities—we don't think it would be straight-forward owing to the physical properties of these micromolar-range ligands. Since our goal was to find agonists, we worry that chasing weak antagonists—if in fact there were any—would be a distraction in a paper that already is data heavy.

4.13. Discussion and results: *"Additionally, in our hands using CB2R knockout mice, at minimum the analgesic effects of '1350 are not due to engagement of CB2Rs" What about the tetrad effects? Analgesic effects were tested in CB1 and CB2 KO mice, but side effects were not tested in these mice.*

We have now included a new experiment testing '1350 in CB1/2 KO mice in the catalepsy test (**Fig. S11D**).

We also tried to get data on the rotarod, but the mice without drug on board were not able to run on the rotarod for the full 300 seconds, despite multiple rounds of training, which limits our ability to draw conclusions from this assay and suggests that there is a motor impairment in the mice themselves that will hinder our ability to confidently draw conclusions about CB2's role. Further, we are not able to redo

the hypothermia experiment because we collected these data in mice implanted with telemetric probes, which we currently do not have access to.

4.14. Methods “The HEK293 clonal cell line (HEK293SL cells) for bioSens-All experiments were derived from HEK293 cells purchased from ATCC.” *The authors should acknowledge the source and validation of cells in the text, since I believe they did not derive them.*

We regret this oversight, and have updated the text to include the reference to the original cell line characterization. We now write (p.51)

“The HEK293 clonal cell line (HEK293SL cells) for bioSens-All experiments were derived and characterized previously(Namkung 2016) from HEK293 cells purchased from ATCC.”

Reporting Summary Text:

“HEK293 clonal cell line (HEK293SL cells) for bioSens-All experiments was derived and characterized in Namkung et al., Nat Commun. 2016 Jul 11;7:12178. Source of original HEK293 cells was American Type Culture Collection (ATCC). HEK293 cell lines are maintained by the supplier. No additional authentication was performed by the authors of this study. Cells were tested for mycoplasma contamination on a regular basis. Cells were free of contaminations.”

Reviewer 5

This is a comprehensive research paper; the authors demonstrate a full picture of preclinical drug discovery targeting CB1 receptors to develop new chemotypes with better therapeutic windows and fewer adverse effects... This study advances our understanding of the therapeutic profiles of CB1R agonists for pain treatment. However, it is very hard to read the manuscript due to the lack of context information and clear writing flow and I recommend a major revision of the text.

The reviewer had several kind things to say about this study, we thank them for their support. They also gave examples of where we can clarify it. We take each in turn.

5.1. *Although '1350 is almost equipotent to CB1R and CB2R in pharmacology assays, the knockout animal revealed its analgetic effect mediated via CB1R but not CB2R. This is an interesting finding, suggesting the different participation of cannabinoid agonists in pain physiology. However, neither the introduction nor the discussion provides any background information about the different roles or profiles of two CB receptors, or the rationale for assessing the selectivity.*

This is well-taken. We have added more text discussing the differences of CB1 and CB2 receptors to the Introduction, and reprised the logic for testing both by knockout in the Results.

Introduction (p.3-4)

“The cannabinoid-1 and -2 receptors (CB1R and CB2R), both members of the lipid family of G protein coupled receptors (GPCRs), are the primary mediators of cannabinoid activity¹⁴. These related receptors are largely differentiated by their expression profiles, with CB1R being expressed throughout the nervous system¹⁵ and body, and CB2R primarily expressed in peripheral immune cells¹⁶, though the exact distribution of the latter is still a subject of debate¹⁷. Based on these expression profiles, and supported by animal studies¹⁸⁻²⁰, CB1R is thought to be the major target involved in the psychotropic and tetrad effects of cannabinoids, as well as their analgesic effects in many tests of nociception²¹, though because of the high similarity of the two receptors, and the peripheral distribution of CB2R, a role for the latter receptor often cannot be discounted without direct testing.”

Results (p.19)

“Because of the high sequence similarity of the CB1 and CB2 receptors, and the potential role of the latter in analgesia, we investigated the role of the two receptors in the analgesia of the docking-derived compounds. Consistent with CB1R being the target of ‘**1350** *in vivo*, total knockout of CB1R in the mouse completely blocked the analgesic effect of ‘**1350**, but not of morphine, in the tail flick assay (**Fig. 5F**).”

5.2. In line 208-211: “The trifluoromethyl group is complemented by van der Waals and quadrupole interactions with residues W2795.43 and T1973.33, as anticipated by the docked structure, and consistent with the improvement in affinity by -1.7 kcal/mol (17-fold in K_i) on its replacement of the original fluorine.”

I assumed the “-1.7kcal/mol” originates from the binding free energy calculation based on docking results, but I did not find any corresponding method description or raw data comparison between ‘1350 (trifluoromethyl) and ‘51486 (fluorine).

These are experimental differences in free energy based on the measured K_i values. ‘51486 (fluorine-containing) has a K_i of 731 nM ($\Delta G_b = 8.4$ kcal/mol) and ‘**60154** (only change is a FluorineCF₃ substitution) has a K_i of 44 nM ($\Delta G = 10.0$ kcal/mol; 17-fold better, $\Delta\Delta G = 1.7$ kcal/mol). We used the values for ‘**60154** for this calculation because ‘**1350** has multiple changes from ‘**51486** that comparing their K_i s directly does not specifically address the F1  CF₃ substitution. We have added text to clarify this point:

“consistent with the improvement in affinity by -1.7 kcal/mol (‘**51486** $K_i = 731$ nM vs. ‘**60154** $K_i = 44$ nM, 17-fold increased K_i from CF₃ addition only) on its replacement of the original fluorine.”

5.3. In line 164-167: “51486, the lipophilic ligand efficiency (LLE) of ‘1350 improved from 3.1 to 4.7 (Fig.2B), whereas ‘496_’s LLE stayed approximately the same (3.2); both compare favorably to an LLE of 2.6 for the positive control CP-55,940, which is substantially more hydrophobic than either of the two new agonists.” *The authors assumed the reader understood the calculation of $LLE = pIC50 - clog P$. However, it seems the $clogP$ value of CP-55,940 is not provided in this paper using the consistent method used for the two new leads, but only acclaims the more hydrophobicity of CP-55,940. I suggest authors to describe the concept of LLE in the method or figure legend and include $cLogP$ value of CP-55,940 in the text or figures.*

Great idea. We now include the definition of LLE directly in the text, as well as the $cLogP$ of CP55,940 (p. 12), as well as in a new **Supplementary Table 13**.

5.4. *In pharmacology assessment, Fig S5, 2-AG was used as positive control? But this information was not mentioned in method or anywhere.*

We used CP-55,940 as the positive control here, and in all our other assays. We were interested in the profile of 2-AG, as it is an endogenous agonist. However, 2-AG was difficult to work with and we did not know how it would respond in these assays, as it is often broken down too quickly to be used in assays with long incubation times. We ultimately stuck with comparisons to CP-55,940 as that was the control used in all our other assays, and widely in the field. Based on this comment and one from Reviewer 4 that the presence of the 2-AG data distracts from the main comparisons of ‘1350 and ‘4042 to CP-55,940, the control ligand used in the *in vivo* studies and widely in the field. We thus removed the 2-AG from the manuscript, hopefully improving clarity.

Fig S6, the compound labels are not clear, such as ‘3737 a-d; and ‘7019-31/29; do this mean different batch of synthesis, please clarify this information.

We regret the confusion. 7019-31, 7800-29, and ‘3737a-d are internal IDs for the compounds. Specifically, compounds ‘3737a-d are the four pure diastereomers of compound ‘3737 (SMILES can be found in Supplementary Table 10). We have updated Fig S6 to have their correct/formal IDs (‘5424, ‘5463, ‘5490, ‘1651, ‘0430, and ‘3386) which correspond to their data and structures in Supplementary Tables 2-3 and 14 and the chemical characterization data.

5.5. *In addition, cryo-EM structure of ‘1350 was used to confirm the docking pose. However, in consistent with bad local resolution (Fig.S8B), the density for ECL2 and other regions surrounding the ligand is fragmented and ambiguous, not enough to determine the rotamer of some residues. These undetermined sidechains include H178, S383, S173 and the former two are key interactions that the authors acclaimed in the text and figure 3. After looking through the method, I strongly suggest that authors ... improve the EM map quality as they gained a reasonable number of good particles. Here are several usual ways may help.*

1. Try polishing the final particle stacks on relion/5.0 and import to cryosparc for final 3D refinement.
2. Try local refinement in cryosparc or relion using masks only cover the receptor region or the receptor-Gi region; the similar approach was used in CB1 receptor structure(8GHV).
3. Try the latest relion/5.0 of 3D refinement with Blush regularisation.

I fully understand that maybe the time would not allow authors to optimise the map quality. In such a case, I suggested authors mention the resolution limitation and rephrase some words more cautiously. For example, In line 206-208, “The major interactions with CB1R predicted by the docking are preserved in the experimental structure, including the key hydrogen-bond between the amide carbonyl of the ligand and S3837.39” maybe rephrase to “despite the local resolution limit that prevents unambiguously modelling all rotamers,are likely to be preserved.”

These suggestions are very helpful, and we have revisited refinement of the map and the model. We had initially tried using a mask that covered only receptor-Gi, but it did not improve resolution. However, and per their suggestion, we were able to use a mask that covers only the receptor. This led to better density for the ligand and the residues around it. We have included this in the manuscript with updated figures and an updated Methods section. We now write (p.14)

“Despite the local resolution limit that prevents unambiguously modelling all rotamers, the density suggests that major interactions with CB1R predicted by the docking are likely preserved in the experimental structure, including the key hydrogen-bond between the amide carbonyl of the ligand and S383^{7,39}.”

5.6. *Furthermore, the density display in Fig.3A is ambiguous at too low counter level. Please refer to the Fig S2D in the published CB1R structure (6N4B...) to present the ligand density precisely.*

We agree. We have increased the contour level around the ligand to better display the density akin to Fig. S2D in the 6N4B paper. Fig. 3A has been updated accordingly.

5.7. *In addition, the ligand of 6N4B (MDMB-Fubinaca) was used to set up the molecular docking, and MDMB-Fub also has similar methyl ester moiety and binding pose to “1350. 6N4B is also used as the initial template for modelling ‘1350 structure, so I suggested author make a supplementary figure to compare the ‘1350 structure and the published 6N4B.*

We now include a new **Figure S9** depicting the similarities in the binding pockets between 6N4B and our newly acquired structure (9DGI).

5.8. *Other minor corrections:*

1. *In line 176: “The one exception is ‘4396, whose Emax of 65% is more consistent with strong partial agonism” I reckon it’s a typo for the compound ‘4936.*

Fixed, thank you.

2. *The figure legend for S4: panel A. “Competition binding data for primary hits and a subset of their analogs at hCB1.” This is the binding assay using rat brain so it should be rCB1 not hCB1.*

Fixed, thank you. New text is included below.

“Figure S4. CB1 binding and functional data for analogs. A. Competition binding data for primary hits and a subset of their analogs at rCB1.”

Reviewer 1

The authors have satisfactorily commented on the points raised and included additional data in many cases. However, the authors response to the below are not satisfactory:

1.7 *The synthesis section has been improved and yields reported, yet it is not clear on what scale each reaction leading to the product was run giving the respective product yields? Are these run on a 100 mg scale for each reaction shown in general methods section or the mmol/mol different in each case? Even if it is run on a 100 mg scale the mmol values for each reactant will be different. This is generally required to understand for the scalability of the methods vis-a vis the reported yields.*

We did, as the Reviewer suggests, use the general synthesis methods described by Methods 1-6 as indicated for each molecule. These methods describe the parallel chemistry approaches employed at Enamine. As such, everything was done on a 100 mg scale with the indicated molar equivalents for the other reactants. To address the Reviewer's concern that for each individual reaction the mol equivalents will necessarily be different amount of mols for each reactant, we have updated the supplemental information file to include further detail of each reaction. We hope this makes the work more reproducible for readers.

1.8 *Rotation values for enantiomers need to be reported. How was the stereochemistry assigned, is it based on cryoEM data?*

We have updated the supplemental information file and associated Methods section to indicate the rotation values of Z8526711350 and Z8526708690, where Z8526711350 was found to be the (-) enantiomer and Z8526708690 was found to be the (+) enantiomer.

Z8526711350 $[\alpha]_D^{21} = -29.1$ (c 1.0, CH₃OH).

Z8526708690 $[\alpha]_D^{21} = 27.7$ (c 2.0, CH₃OH).

The stereochemistry was assigned based on the cryoEM data.

Reviewer 2

I am satisfied with the revisions. My criticism points were sufficiently addressed.

Thank you!

Reviewer 4

This is a revision of a previously reviewed study. The authors have generally addressed our prior comments and concerns well, incorporating new data and making appropriate revisions to the text. However, a few points still warrant further consideration.

4.10: *We apologize for not clearly conveying our thoughts on functional selectivity (bias). Our intention was to emphasize caution when calculating and interpreting bias using E_{max}/EC_{50} values and relative activity, as these parameters are not equally weighted, as EC_{50} influence the relative activity more than E_{max} which is limited in span ... While this may not constitute statistical bias, compound 1350 shows a clear reduction in E_{max} for G13 and, and to some extent β -arrestin, which could have some physiological importance. Our comment was more editorial, as this is the standard way biases are reported in the literature, which we are also bound to use in our many studies, but clearly imperfect and can lead to misinterpretation by the reader.*

We appreciate the kind tone of this comment; it is well-taken. In the Discussion, we now write that the G_{13} difference was the most notable, that its physiological effect is not currently understood, and that this effect merits further exploration. In the Results, we mention that ‘1350 is relatively more efficacious at recruiting $G_{i/o}$ and G_{13} . To further strengthen this point, we now also write:

Notably, ‘1350’s E_{max} for G_{13} and β -arrestin-2 was reduced, suggestive of partial rather than full agonism of these pathways (Fig. S5G), which may have some physiological relevance....

We have toned down the concluding statement so as not to mislead reader:

In summary, ‘1350 shows no strong functional selectivity (“bias”) but is both more potent and relatively more efficacious than CP-55,940 at CB1R but not CB2R.

4.11: We regret any confusion regarding our comment on the lack of positive controls for G_s and G_q signaling. Our concern was not about testing other ligands for CB pathways showing coupling to G_s and G_q , but rather the absence of controls validating the assay with known G_s and G_q receptors.

This too is well-taken. We now include two supplemental figure panels showing G_q and G_s control experiments with PTHR stimulated by PTH and GSHR stimulated by Ghrelin, respectively (Fig. S5D), supporting assay validation. This technology has been tested and published on 100 GPCRs with diverse coupling partners, and cited in the Method section of the manuscript (Avet, C. *et al.* Effector membrane translocation biosensors reveal G protein and β arrestin coupling profiles of 100 therapeutically relevant GPCRs. *Elife* 11, e74101 (2022)).

Reviewer 5

Very glad to see the density improvement for the ligand and interactive residues in the new local refinement cryo-EM maps. Well done! Current pdb validation report shows only the focus map and receptor regional model has been deposited. Please deposit both the local refinement map (receptor-focused as additional map) and the previous consensus CB1-Gi complex map into your pdb deposition... and merge the new receptor model into your complex model and deposit the intact complex.

Except from the above small comment on structural data deposition, current version addressed all my concerns and I'm also pleasant to accept this manuscript to publish.

We thank the reviewer for the support of the manuscript. We have newly uploaded a composite CB1-Gi map and model per the reviewer’s request.

PS the last tiny thing to fix is in the Fig 2D, where the molecular profile of 4936 was wrongly pasted.

We regret this mistake, it has been fixed in the current draft.